# CODA-BENCH: Can Code Agents Handle Data-Intensive Tasks?

**Yuxin Zhang** [1]  **Ju Fan** [1]  **Meihao Fan** [1]  **Shaolei Zhang** [1]  **Xiaoyong Du** [1]

## Abstract

Advanced agents are increasingly demonstrating the potential to operate as autonomous engineers, creating a growing demand for evaluation benchmarks that capture the complexity of real-world development. Such environments typically involve both complex code and large-scale data (i.e., file system). However, existing benchmarks usually evaluate code-centric or data-centric capabilities in isolation, leaving a clear gap with real development scenarios. In this paper, we bridge this gap by introducing CODA-BENCH, the first benchmark to jointly evaluate code and data intelligence in a data-intensive environment. We construct a data-intensive Linux sandbox based on the Kaggle ecosystem (containing hundreds of datasets), where agents must actively explore complex file hierarchies to identify relevant resources and generate code for data-driven analytical tasks. CODA-BENCH comprises 1,009 tasks spanning 31 communities, with each task environment containing an average of 980 files, simulating realistic data scale and noise. Evaluations of advanced agents reveal that even top-performing systems struggle to effectively integrate data discovery with code execution, achieving a success rate of only 61.1%. These results highlight a substantial gap in current agentic capabilities for data-intensive tasks and point to promising directions for future research[*].

## 1. Introduction

Large language models (LLMs) have evolved from conversational assistants into autonomous agents capable of executing complex workflows (OpenAI et al., 2024; Wei

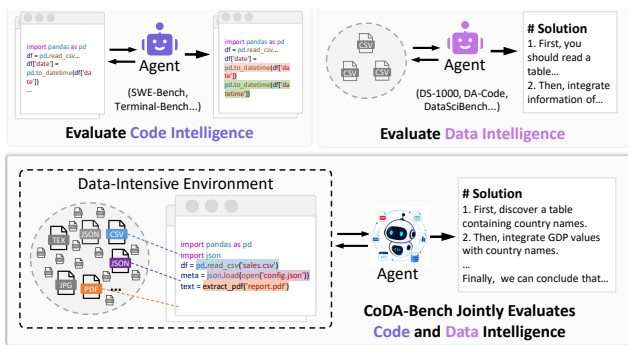

*Figure 1.* CODA-BENCH assesses an agent's capacity to leverage code to solve complex problems in data-intensive environments.

et al., 2022). This shift is especially pronounced in software development, giving rise to tools like Claude Code, Cursor, and Codex CLI that function as autonomous engineers (Jimenez et al., 2024; Liu et al., 2025). As these agents become integrated into professional workflows, rigorous evaluation of their real-world capabilities becomes essential (Wang et al., 2024; Xi et al., 2023).

In real-world deployments, the value of an autonomous agent hinges on interacting with large-scale data in file systems, going beyond solving isolated algorithmic problems (Yang et al., 2024a). An ideal agent should navigate directory hierarchies, identify relevant files from hundreds of candidates, and perform appropriate operations without requiring users to specify targets (Hong et al., 2024; Wu et al., 2023). This capability requires dual intelligence: *Code Intelligence*, which enables agents to generate syntactically correct and logically sound programs (Guo et al., 2024; Lozhkov et al., 2024); and *Data Intelligence*, which allows agents to locate and leverage correct information sources in complex data landscapes (Zhang et al., 2025b). A critical question thus arises: **Do current state-of-the-art code agents integrate both code and data intelligence to handle data-intensive tasks?**

Existing benchmarks typically evaluate code intelligence and data intelligence in isolation, failing to assess their coupled capabilities. Code-centric benchmarks focus on code correctness or repository-level maintenance (Chen et al., 2021a; Austin et al., 2021; Zhuo et al., 2025; Jimenez et al., 2024; Xie et al., 2024), yet largely ignore challenges intro-

[1]Renmin University of China. Correspondence to: Shaolei Zhang (Corresponding Author) <zhangshaolei98@ruc.edu.cn>.

*Proceedings of the 43rd International Conference on Machine Learning*, Seoul, South Korea. PMLR 306, 2026. Copyright 2026 by the author(s).

[*] Project: https://coda-bench.github.io/
Code: https://github.com/ruc-datalab/CoDA-Bench
Data: https://huggingface.co/datasets/RUC-DataLab/CoDA-Bench

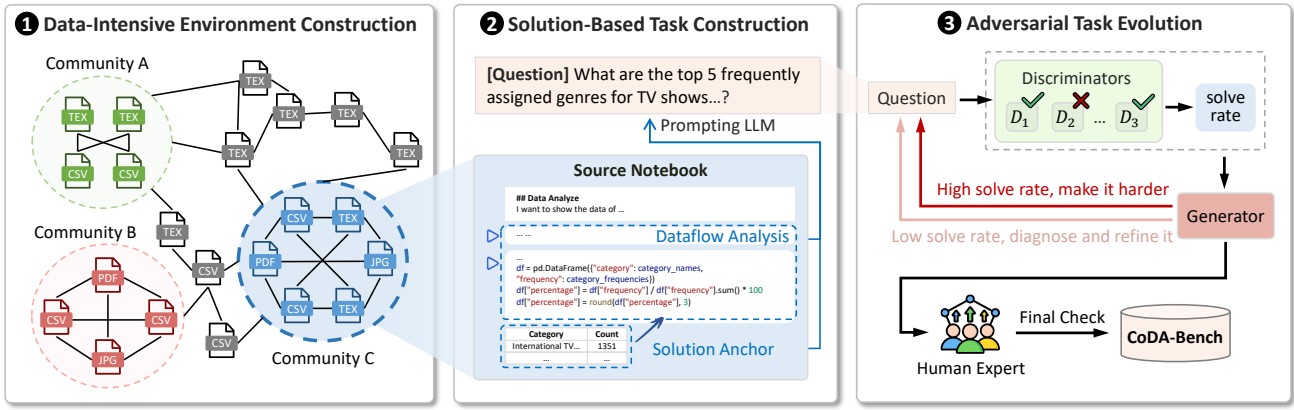

*Figure 2.* Construction method of CODA-BENCH. We construct semantically coherent environments using dataset co-occurrence graphs, extract tasks with closed-form answers from real Kaggle notebooks, and verify quality through adversarial evaluation.

duced by massive, heterogeneous data in real-world settings. Conversely, data-centric benchmarks assess whether agents can process given data through code, but rely on standalone Python scripts and overlook the need to discover and access large-scale data within shell-based environments (Lai et al., 2023; Huang et al., 2024b; Egg et al., 2025). This isolated evaluation paradigm creates a gap between benchmark performance and real-world utility, where data is rarely presented on a silver platter. An agent unable to navigate complex data environments renders even advanced coding capabilities ineffective. These limitations highlight the urgent need for a benchmark that jointly measures both code and data intelligence.

To bridge this gap, we introduce CODA-BENCH (**Co**de and **Da**ta-intensive **Bench**mark), the first benchmark to jointly evaluate the code and data intelligence of agents. Constructing such a realistic benchmark is non-trivial, as randomly generated files are trivially distinguishable from target data, whereas manually curating hundreds of related files is unscalable. Fortunately, the long-standing data science community provides an ideal setting. We leverage the Kaggle ecosystem, which contains interconnected datasets and human-written solution code, to construct our benchmark. Specifically, we curate large-scale data sources from Kaggle and establish a data network by analyzing natural co-occurrence patterns within human workflows. We then propose a scalable and verifiable task construction framework to generate data-intensive analytical tasks. Finally, agents are placed in a data-intensive Linux sandbox where they must incrementally explore data and develop code to complete tasks. Figure 1 illustrates the evaluation paradigm of CODA-BENCH.

CODA-BENCH comprises 1,009 tasks spanning 31 data communities, with evaluation environments averaging 980 files each. Evaluation of state-of-the-art agents (Codex CLI, Claude Code, and Openhands) reveals clear limitations:

even top-performing models achieve only 61.1% execution accuracy on CODA-BENCH and 49.6% on CODA-HARD, a more challenging subset. Further analysis indicates that current code agents fall far short of autonomously completing data-intensive tasks, leaving substantial room for improvement.

## 2. Related Work

**Code-centric Benchmarks.** Evaluation of LLMs for code generation has progressed from relatively simple function-level tasks toward more challenging settings that reflect realistic software development. Early efforts primarily measured functional correctness by executing unit tests on generated programs (Chen et al., 2021a; Austin et al., 2021; Hendrycks et al., 2021; Du et al., 2024; Zhuo et al., 2025; Jain et al., 2025; Liu et al., 2024a; Cassano et al., 2023; Chen et al., 2025). Recent work increasingly evaluates agents in realistic software engineering workflows. One line of research focuses on issue-driven code repair in real repositories, such as SWE-bench (Jimenez et al., 2024) and its variants (Yang et al., 2024b; Zan et al., 2026). Meanwhile, interactive environment benchmarks assess agent capabilities in web, desktop, and terminal-level interactions (Zhou et al., 2024; Xie et al., 2024; Deng et al., 2023; Liu et al., 2024b; Merrill et al., 2026). While these benchmarks advance the evaluation of agents on real-world tasks, they generally assume that all data required to complete the task is already prepared and readily available (i.e., external data is explicitly provided in the environment), overlooking the fact that agents must first discover valuable information in complex data environments by themselves during real-world development.

**Data-centric Benchmarks.** Understanding and manipulating data are essential capabilities for intelligent agents. Early benchmarks primarily evaluated LLMs' abilities to understand structured data (Pasupat & Liang, 2015; Chen et al.,

2020; 2021b;c; Zhao et al., 2022; Nan et al., 2022; Cheng et al., 2022; Qiu et al., 2024) and generate code for data processing (Yu et al., 2018; Li et al., 2024; Ouyang et al., 2026; Hu et al., 2024). More recent efforts have shifted toward assessing agents' competence in solving complex, end-to-end data science tasks across a broader spectrum, such as DA-Code (Huang et al., 2024b), DABstep (Egg et al., 2025), KramaBench (Lai et al., 2026), DataSciBench (Zhang et al., 2025a), DAComp (Lei et al., 2026), ScienceAgentBench (Chen et al., 2025), and DiscoveryBench (Majumder et al., 2025). Despite covering diverse data science scenarios of data wrangling, machine learning, and exploratory data analysis, these benchmarks share a common limitation: all relevant data files are explicitly provided to the agent. Moreover, most of them emphasize relatively simple operations (such as code generation) and rarely require agents to interact with large-scale datasets through realistic environments like the terminal.

## 3. Benchmark Construction

In this paper, we introduce CODA-BENCH, a benchmark designed to jointly evaluate the code intelligence and data intelligence of agents, thereby assessing whether agents can accomplish complex tasks through code in data-intensive environments. Building such a realistic and verifiable benchmark poses three key challenges: (1) creating realistic data environments that require genuine data discovery capabilities, (2) collecting tasks that reflect authentic real-world needs while still permitting objective evaluation, and (3) ensuring task quality through systematic verification. To address these challenges, we propose a scalable framework for benchmark construction (Figure 2), described as follows.

### 3.1. Data-Intensive Environment Construction

The primary objective of CODA-BENCH is to assess whether code agents can discover task-relevant data within large collections of semantically similar files and then perform subsequent operations. A naive strategy of filling environments with randomly generated files fails to reflect real-world difficulty, as such files are easily distinguished from target data based on superficial features. Realistic evaluation requires *in-distribution* noise, where distractor files share topical and structural characteristics with the target data while remaining irrelevant to the task.

To construct challenging environments at scale without prohibitive manual curation, we leverage the Kaggle ecosystem[2] which hosts over 646,615 publicly available datasets across diverse domains and also offers human-authored notebooks that tackle complex analytical problems. When analysts write notebooks, they deliberately explore topically

[2] https://www.kaggle.com

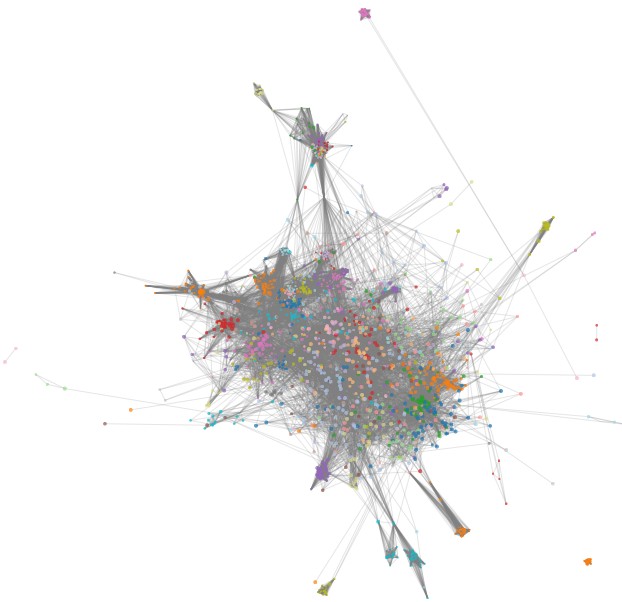

*Figure 3.* Dataset co-occurrence network showing 21,122 Kaggle datasets (nodes) and their co-usage relationships (edges). Node size indicates usage frequency; colors represent communities detected by the Leiden algorithm.

related data, creating implicit associations among semantically similar data sources. These associations provide a principled basis for determining which data should co-occur in realistic evaluation environments.

**Graph-based Data Relationship Modeling.** To construct a realistic data environment, we aim to build a massive relational network across hundreds of Kaggle datasets. Specifically, we propose graph-based data relationship modeling to capture semantic relationships through co-occurrence patterns in Kaggle notebooks. Let $\mathcal{D} = \{d_1, \ldots, d_n\}$ denote all data and $\mathcal{N} = \{n_1, \ldots, n_m\}$ denote all notebooks, where each notebook $n_j$ references a subset $\mathcal{D}_j \subseteq \mathcal{D}$. We construct an undirected weighted graph $G = (\mathcal{D}, E, w)$ with edge weights defined by co-occurrence frequency:

$$w(d_i, d_k) = \sum_{j=1}^{m} \mathbb{1}\left[d_i \in \mathcal{D}_j \wedge d_k \in \mathcal{D}_j\right], \quad (1)$$

where $\mathbb{1}[\cdot]$ is the indicator function. An edge $e_{ik} \in E$ exists if and only if $w(d_i, d_k) > 0$.

**Community Partitioning.** The raw co-occurrence graph encompasses heterogeneous domains. To obtain semantically coherent environments, we partition the graph into domain-specific communities using the Leiden algorithm (Traag et al., 2019) with resolution parameter $\gamma = 1.0$. This yields $|\mathcal{C}|$ distinct communities $\mathcal{C} = \{C_1, \ldots, C_{|\mathcal{C}|}\}$, each containing data that share domain characteristics.

For each task associated with target data $\mathcal{D}^* \subset C_k$, we construct the data-intensive evaluation environment by in-

cluding all data within community $C_k$. The data in $C_k \setminus \mathcal{D}^*$ serve as in-distribution distractors sharing topical similarity with the target data. This design ensures that agents cannot rely on superficial keyword matching or format-based filtering, but must perform fine-grained semantic reasoning to identify appropriate data sources. Finally, each environment contains 980 data instances in multiple formats, including CSV, JSON, Parquet, images, and PDFs, with total sizes ranging from 20.3 MB to 45.4 GB. Appendix B provides a complete description of the graph construction procedure and visualizes the resulting co-occurrence graph.

### 3.2. Solution-Based Task Construction

Given the data-intensive environments described above, we then address the challenge of constructing verifiable tasks that reflect authentic needs. Kaggle notebooks document complete solutions to data analysis problems along with numerical results, providing an ideal foundation for task construction. We propose *solution-based back-construction*, a methodology that derives benchmark tasks from verified solutions in Kaggle notebooks.

We refer to the precise numerical results produced in Kaggle notebook solutions as solution anchors, which include statistics, rankings, correlations, and aggregations that domain experts consider meaningful enough to report. Such anchors are deterministically reproducible and verifiable given the same data and computational procedures, and they reflect authentic questions that practitioners genuinely care about. Accordingly, we propose solution-based back-construction, which works backward from these anchors to reconstruct the questions that originally motivated their solution.

**Anchor Identification.** We parse each notebook to identify anchors from cell outputs. Specifically, we employ a combination of static analysis and dynamic verification to detect numerical outputs that can serve as solution anchors. For static analysis, we leverage an advanced LLM to select outputs that are both verifiable and non-trivial as candidate anchors. We then perform dynamic verification on these candidates via solution path reconstruction. We trace the provenance of each candidate anchor through static dataflow analysis. For an anchor $a$ produced in cell $c_t$, we identify the minimal set of input files $\mathcal{D}_a \subseteq \mathcal{D}_n$ and the sequence of transformations $\mathcal{T}_a = \langle \tau_1, \tau_2, \ldots, \tau_k \rangle$ required to compute $a$. We verify answer uniqueness by re-executing the extracted computation path and confirming that the reproduced result matches the original anchor within numerical tolerance $\varepsilon = 10^{-6}$.

**Question Formulation.** Given a selected solution anchor, we employ an LLM to generate natural language questions based on the resulting output and the reconstructed solution path. We require each question to specify the task goal clearly while avoiding any disclosure of the underlying so-

lution pathway. All generated questions are subsequently reviewed by human annotators to eliminate ambiguous cases or questions that admit multiple valid interpretations. Appendix C provides detailed examples.

### 3.3. Adversarial Task Evolution and Verification

The solution-based back-construction ensures task correctness, but initial tasks may not sufficiently challenge state-of-the-art agents. We aim to evolve tasks toward maximal difficulty while preserving solvability. These goals create an inherent tension, as increasing difficulty risks introducing ambiguity or insufficient information, whereas ensuring solvability may result in trivial tasks. To navigate this trade-off, we propose an *adversarial evolution* framework.

**Adversarial Evolution Framework.** Inspired by generative adversarial networks (Goodfellow et al., 2020), we formulate task evolution as a two-player game between a *generator* $G$ that maximizes task difficulty and a *discriminator* $F$ that attempts to solve any given task. Let $q$ denote a task instance with ground-truth answer $a_q$. The adversarial objective can be expressed as:

$$\min_G \max_F \mathcal{L}(G, F) = \mathbb{E}_{q \sim G} \left[ \mathbb{1}[F(q) = a_q] \right] \quad (2)$$

Unlike standard GANs, our framework employs state-of-the-art LLMs as both generator and discriminator, with discrete task modifications replacing continuous parameter updates.

**Iterative Refinement Process.** We instantiate this adversarial game through iterative refinement. At each iteration $t$, the generator $G$ produces a modified task $q^{(t)}$ from the previous version $q^{(t-1)}$. To prevent overfitting to any single LLM, the discriminator comprises an ensemble of $K$ models $\{F_1, F_2, \ldots, F_K\}$ randomly sampled from a pool. The ensemble computes the solve rate as:

$$r^{(t)} = \frac{1}{K} \sum_{k=1}^{K} \mathbb{1}[F_k(q^{(t)}) = a_q]. \quad (3)$$

The generator produces the next iteration based on $r^{(t)}$. When the solve rate exceeds a predefined threshold, the task is deemed insufficiently challenging. The generator then examines successful solution trajectories to identify opportunities for increasing difficulty. Conversely, when the solve rate falls below the threshold, the generator performs diagnostic analysis on failure trajectories to determine whether failures stem from genuine difficulty or from task defects such as ambiguous wording, missing information, or non-unique answers. If defects are identified, the generator refines the task accordingly. If failures reflect inherent difficulty and the task remains solvable, the task proceeds to human verification. Once reviewers confirm solvability, the iteration terminates and the task is accepted into the

benchmark. Appendix C provides the complete pseudocode and a detailed worked example illustrating the adversarial evolution process.

Using the above approach, we construct CODA-BENCH based on large-scale data from the Kaggle ecosystem and human-written code in notebooks. CODA-BENCH requires agents to solve data-intensive tasks through code, thereby jointly evaluating the code intelligence and data intelligence of agents. More importantly, the entire construction method is scalable and can be used to build datasets at large scale.

### 3.4. Pipeline Statistics

Table 1 summarizes the filtering process across all construction stages. Starting from 323 dataset communities in the Kaggle co-occurrence graph, the pipeline progressively filters candidates through environment construction, task extraction, and quality verification stages, ultimately producing 1,009 high-quality tasks. The overall pass rate of 72.3% (from question generation to final benchmark) demonstrates effective quality control while maintaining reasonable data efficiency.

*Table 1.* Benchmark construction pipeline statistics. Each stage progressively filters candidates to ensure quality.

| Stage | Count | Pass Rate |
|---|---|---|
| *Environment Construction (§3.1)* | | |
| Dataset communities | 323 | – |
| DA communities selected | 31 | 9.6% |
| Notebooks collected | 30,624 | – |
| Notebooks selected | 1,361 | 4.4% |
| *Task Construction (§3.2)* | | |
| Solution anchors extracted | 1,395 | – |
| Candidate questions generated | 1,395 | – |
| *Adversarial Evolution & Verification (§3.3)* | | |
| Questions passing evolution directly | 742 | – |
| Questions revised & passed evolution | 411 | – |
| Questions rejected during evolution | 242 | – |
| Total entering human verification | 1,153 | 82.7% |
| Tasks passing human verification | 1,009 | 87.5% |
| **Overall** | **1,009** | **72.3%** |

## 4. CODA-BENCH

Following the above approach, we finally build CODA-BENCH, the evaluation protocol and statistics are introduced below.

### 4.1. Task Definition

Each task simulates a realistic data analysis scenario in which an agent operates autonomously within an isolated sandbox environment. The sandbox is a Linux environment with a file system containing hundreds of data files. The agent starts at the root directory and receives only a natural-language instruction describing the analytical objective (e.g., "What are the top 5 most frequently assigned genres for TV shows and their counts?"). It must complete the task without prior information about file locations, filenames, or data schemas, requiring autonomous exploration and discovery of relevant data.

Formally, a task is defined as a tuple $\mathcal{T} = (q, \mathcal{F}, a^*)$, where $q$ denotes the natural language instruction, $\mathcal{F} = \mathcal{F}_{\text{target}} \cup \mathcal{F}_{\text{distractor}}$ represents the file system containing both target and distractor files, and $a^*$ is the ground-truth answer. To complete a task, the agent must explore the file system, identify relevant files among semantically similar distractors, comprehend file structures across diverse formats, and write code to derive the final answer. Appendix D illustrates a complete task instance with an example solution.

### 4.2. Evaluation Metrics

CODA-BENCH evaluates agents along two dimensions corresponding to data intelligence and code intelligence.

**Discovery Accuracy (DA)** measures data intelligence, i.e., the ability to locate relevant data sources within complex file systems. Let $\mathcal{F}_{\text{used}}^{(t)}$ denote the files accessed in the agent's solution and $\mathcal{F}_{\text{target}}^{(t)}$ denote the ground-truth target files for task $t$. Discovery Accuracy computes the proportion of tasks where the agent successfully identifies all of the required data:

$$\text{DA} = \frac{1}{|\mathcal{T}|} \sum_{t \in \mathcal{T}} \mathbb{1} \left[ \mathcal{F}_{\text{used}}^{(t)} = \mathcal{F}_{\text{target}}^{(t)} \right]. \tag{4}$$

**Execution Accuracy (EA)** measures code intelligence, i.e., the ability to write correct programs that derive accurate answers. Given the agent's output $a_t$ and ground-truth $a_t^*$, Execution Accuracy computes the proportion of tasks with correct answers after normalization (including rounding, whitespace removal, and case standardization):

$$\text{EA} = \frac{1}{|\mathcal{T}|} \sum_{t \in \mathcal{T}} \mathbb{1} \left[ \text{normalize}(a_t) = \text{normalize}(a_t^*) \right]. \tag{5}$$

Together, these metrics provide comprehensive assessment of agent capabilities. DA isolates data discovery performance independent of downstream computation, while EA captures end-to-end task completion requiring both accurate data discovery and correct code execution.

### 4.3. Benchmark Statistics

CODA-BENCH comprises 1,009 tasks across 31 semantically coherent communities derived from our graph-based partitioning. The environments range from 10 to 8,158 files,

*Table 2.* Position of CODA-BENCH among the existing benchmarks.

| Benchmark | Capability | Environments | | Tools | | Tasks | | #Tasks |
|---|---|---|---|---|---|---|---|---|
| | | Data Scale | Unused Data | Code | Terminal | Source | w/o Guidance | |
| **GAIA** (Mialon et al., 2023) | General | Multi-file (≤10) | ✗ | ✗ | ✗ | Human QA | ✓ | 466 |
| **HLE** (Center for AI Safety et al., 2026) | | Multi-file (≤10) | ✗ | ✗ | ✗ | Exams | ✓ | 3,000 |
| **SWE-Bench** (Jimenez et al., 2024) | Software | Repo-level | ✗ | ✓ | ✓ | GitHub | ✓ | 2,294 |
| **Terminal-Bench** (Merrill et al., 2026) | Engineering | Multi-file (≤10) | ✗ | ✓ | ✓ | Synthetic | ✓ | 89 |
| **MLE-bench** (Chan et al., 2024) | Machine | Multi-file (≤10) | ✗ | ✓ | ✗ | Kaggle | ✓ | 75 |
| **MLAgentBench** (Huang et al., 2024a) | Learning | Multi-file (≤10) | ✗ | ✓ | ✗ | Synthetic | ✓ | 9,641 |
| **DS-1000** (Lai et al., 2023) | | 0 | ✗ | ✗ | ✗ | StackOverflow | ✗ | 1,000 |
| **DA-Code** (Huang et al., 2024b) | | 1 | ✗ | ✗ | ✗ | Tutorials | ✗ | 500 |
| **DSBench** (Jing et al., 2025) | | Multi-file (≤10) | ✗ | ✓ | ✗ | Kaggle | ✗ | 540 |
| **DABstep** (Egg et al., 2025) | Data | 7 | ✗ | ✗ | ✗ | Synthetic | ✗ | 450 |
| **DataSciBench** (Zhang et al., 2025a) | Science | Multi-file (≤10) | ✗ | ✗ | ✗ | CodeGeeX | ✗ | 519 |
| **DAComp** (Lei et al., 2026) | | Repo-level | ✗ | ✗ | ✗ | Enterprise | ✗ | 210 |
| **ScienceAgentBench** (Chen et al., 2025) | | Multi-file (≤10) | ✗ | ✗ | ✗ | Scientific papers | ✗ | 102 |
| **DiscoveryBench** (Majumder et al., 2025) | | Multi-file (≤10) | ✗ | ✗ | ✗ | Scientific papers | ✗ | 264 |
| **CODA-BENCH** | Data-Intensive Analysis | Community-level (980) | ✓ | ✓ | ✓ | Kaggle | ✓ | 1,009 |

*Table 3.* Statistics of CODA-BENCH and CODA-HARD. Signal-to-noise ratio (SNR) is defined as the fraction of key files relative to total files in the environment.

| Statistic | CODA-BENCH | CODA-HARD |
|---|---|---|
| Total tasks | 1,009 | 119 |
| Data communities | 31 | 15 |
| Files per environment | 980.8 | 1421.7 |
| Key files per task | 1.3 | 2.4 |
| Signal-to-noise ratio | 0.0105 | 0.0142 |

spanning CSV, JSON, Parquet, PDF, and image formats. We curate CODA-HARD, a subset of 119 tasks that challenge both data intelligence and code intelligence simultaneously.

Tasks qualify for CODA-HARD based on two criteria. First, *Data Complexity* requires discovering at least two target files from a large collection. Second, *Code Complexity* requires that the reference solution exceeds 30 effective lines of code. This filtering yields tasks where agents must integrate information across multiple sources and construct non-trivial programs. Table 3 summarizes key statistics for both benchmarks.

### 4.4. Comparison with Existing Benchmarks

Table 2 illustrates the positioning of CODA-BENCH relative to existing benchmarks. Unlike prior benchmarks, which typically supply only the oracle files strictly necessary for each task, CODA-BENCH is the first to introduce large-scale, relevant yet uncurated data into the evaluation environment. In practical development settings, agents must identify critical information from massive, unstructured data corpora before tackling downstream tasks. CODA-BENCH is explicitly designed to achieve a more realistic assessment of an agent's ability to discover, select, and effectively exploit useful information, thereby narrowing the gap between benchmark evaluations and real-world development scenarios. By jointly evaluating data intelligence and code intelligence, CODA-BENCH offers a more comprehensive measure of agent capability in data-intensive environments.

## 5. Evaluation

### 5.1. Experimental Setup

We evaluate state-of-the-art coding agents on CODA-BENCH to assess their ability to complete tasks in data-intensive environments. For native CLI tools, we test Claude Code[3] with Claude-Opus-4.7[4], Claude-Sonnet-4.6, and Claude-Opus-4.6, and Codex CLI[5] with GPT-5.5[6] under their official default configurations, capturing out-of-the-box performance in realistic deployments. To evaluate the underlying LLMs, we adopt OpenHands (Wang et al., 2025) as a unified agent framework with backbone models including GPT-5.5, Claude-Opus-4.7, Kimi-K2.6 (Team et al., 2026), DeepSeek-V4-Pro (DeepSeek-AI et al., 2025). We also evaluate Mini-SWE-Agent (Yang et al., 2024a), a repository-level agent, with GPT-5.5. All experiments are conducted in isolated sandbox environments with identical computational resources.

### 5.2. Main Results

To understand how current coding agents perform on data-intensive tasks, we evaluate both proprietary and open-weight models across native CLI tools and the framework-based agents. Table 4 reports the complete results.

Introducing large-scale data into the environment presents

---

[3] https://code.claude.com
[4] https://www.anthropic.com/news/claude-4
[5] https://openai.com/codex/
[6] https://openai.com/gpt-5

*Table 4.* **Main results on CODA-BENCH and CODA-HARD.** Comparison of Discovery Accuracy (DA), Execution Accuracy (EA), average trajectory length (turns and tokens), and cost per task. Best results in each column are in **bold**, second-best underlined. Cost for Claude Code is based on built-in billing; other costs are estimated based on OpenRouter pricing. $\sim$ indicates estimated values.

| System | Model | CODA-BENCH | | CODA-HARD | | Avg. Turns | Avg. Tokens | $/Task |
|---|---|---|---|---|---|---|---|---|
| | | DA (%) | EA (%) | DA (%) | EA (%) | | | |
| *Native CLI Tools* | | | | | | | | |
| Codex CLI | GPT-5.5 (xhigh) | 75.0 | 60.5 | 57.0 | 42.9 | 6.8 | 390,992 | $\sim$1.42 |
| Codex CLI | GPT-5.5 | 74.9 | 60.3 | 61.4 | 47.9 | 6.8 | 380,558 | $\sim$1.39 |
| Claude Code | Opus-4.7 | 77.3 | 51.9 | 61.4 | 45.4 | 16.1 | 123,699 | 0.22 |
| Claude Code | Opus-4.6 | 71.3 | 47.8 | **68.4** | 45.4 | 17.2 | 113,433 | 0.20 |
| Claude Code | Sonnet-4.6 | 77.9 | 53.8 | 61.4 | 42.9 | 14.7 | 81,714 | 0.11 |
| *Framework-Based Agents* | | | | | | | | |
| Mini-SWE-agent | GPT-5.5 | **83.0** | **61.1** | 67.5 | **49.6** | 32.5 | 101,017 | $\sim$0.39 |
| OpenHands | GPT-5.5 | 82.1 | 59.7 | 63.2 | 44.5 | 18.1 | 154,913 | $\sim$0.65 |
| OpenHands | Opus-4.7 | 72.0 | 49.3 | 59.6 | 38.7 | 24.4 | 205,684 | $\sim$1.17 |
| OpenHands | Kimi-K2.6 | 71.5 | 43.8 | 59.6 | 37.0 | 39.4 | 380,292 | $\sim$0.41 |
| OpenHands | DeepSeek-V4-Pro | 75.9 | 49.0 | 58.8 | 36.1 | 35.8 | 330,161 | $\sim$0.15 |

substantial challenges for state-of-the-art agents. Among the evaluated systems, Mini-SWE-Agent with GPT-5.5 achieves the highest execution accuracy at 61.1%, followed closely by OpenHands with GPT-5.5 at 59.7%. Although these top-performing agents demonstrate strong coding capabilities on isolated benchmarks, they struggle when required to autonomously identify relevant data sources within large, unstructured datasets. In particular, Discovery Accuracy (DA) evaluates an agent's ability to locate target files among hundreds of candidates. Even the best-performing agents fail to identify the correct data in nearly 20% of tasks, underscoring the difficulty of navigating semantically similar files in our community-based environments.

We also observe that model-framework alignment affects performance. GPT-family models benefits from the Mini-SWE-agent with a 0.8% point improvement in EA over its native CLI, while Claude-family performs better within its native environment (51.9% vs 49.3% in EA). These results indicate that optimal performance requires pairing models with compatible agent scaffolds. Overall, considerable room for improvement remains before agents can autonomously complete complex tasks in data-intensive environments.

### 5.3. Performance on Complex Tasks

To evaluate agent capabilities under more demanding conditions, we analyze performance on CODA-HARD, the subset requiring coordination across multiple files and complex data processing pipelines. All models experience substantial performance degradation on CODA-HARD compared to the full benchmark, confirming that multi-source integration poses fundamental challenges beyond single-file analysis. Among top-performing models, Mini-SWE-Agent achieves 49.6% EA on CODA-HARD, while Claude Code with Opus-4.6 demonstrates strong resilience with 68.4%

DA and 45.4% EA, suggesting effective multi-step reasoning capabilities. In the development of agents toward becoming autonomous engineers, CODA-HARD provides a challenging benchmark.

### 5.4. Cost-Performance Analysis

To examine the trade-offs between performance and computational cost, we evaluate the cost efficiency of different agents. Table 4 reports the average cost per task for each system. For Native CLI Tools, Claude Code demonstrates the most favorable cost-performance ratio, achieving 53.8% EA at $0.11 per task with Sonnet-4.6, representing a significant cost reduction compared to other alternatives. Codex CLI with GPT-5.5 achieves strong performance (60.3% EA) but at significantly higher cost ($1.39 per task). Notably, Sonnet-4.6 consumes significantly fewer tokens (81,714 avg.) compared to Codex CLI (380,558 avg.), demonstrating more efficient tool calling that results in lower operational costs. These results suggest that the degree of optimization varies across different native CLI tools, with each offering distinct trade-offs between cost and performance.

## 6. Analysis

We conduct extensive analyses to understand agent performance on data-intensive tasks.

### 6.1. Challenges Arising from Intensive Data

To disentangle the contributions of data discovery and code generation to overall task difficulty, we conduct an ablation study via providing oracle data on CODA-HARD. We compare two settings. In the **Community** setting, agents must discover relevant files from the full environment. In the **Oracle** setting, we provide exact paths to required files in

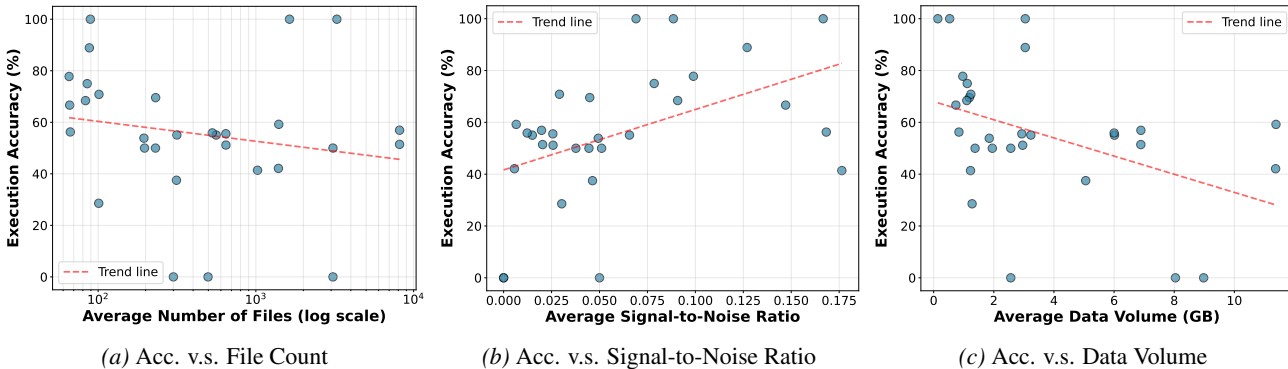

*(a)* Acc. v.s. File Count        *(b)* Acc. v.s. Signal-to-Noise Ratio        *(c)* Acc. v.s. Data Volume

*Figure 4.* Impact of environmental characteristics on GPT-5.5 performance (on Top 30 communities). (a) File count: Spearman $\rho = -0.271$, $p = 0.148$. (b) Signal-to-noise ratio: $\rho = 0.466$, $p < 0.01$, demonstrating community-based construction creates semantically challenging distractors. (c) Data volume: $\rho = -0.461$, $p < 0.01$, indicating I/O bottlenecks at scale.

the task prompt, mimicking benchmarks that assume known data context. Figure 5 presents the results.

We find that data discovery accounts for a substantial share of the overall task difficulty. Supplying oracle data leads to marked performance improvements, confirming that identifying relevant files among thousands of candidates is a genuine challenge. The ablation reveals distinct capability profiles across agents. Claude Code (Sonnet-4.6) improves from 45.4% to 73.1% with oracle data, a gain of 27.7 points, while OpenHands (GPT-5.5) improves from 44.5% to 68.9%, a gain of 24.4 points. These substantial improvements indicate that data discovery constitutes a major bottleneck for current agents in data-intensive environments.

Critically, even with oracle context, agents achieve only 71.0% average accuracy. The remaining 29.0% failure rate demonstrates that CODA-HARD poses substantial challenges for code generation, including multi-source integration across heterogeneous schemas, semantic ambiguity that requires domain knowledge, and complex multi-step reasoning. These challenges persist even when the correct files are explicitly provided to the agent. Overall, CODA-BENCH is the first benchmark to unify data discovery and code generation within a single framework, introducing challenges that reflect capabilities essential for real-world development and providing a valuable evaluation platform for the future advancement of agents.

### 6.2. Impact of Data-Intensive Environments

To investigate how data-intensive environments challenge coding agents, we analyze the correlation between performance and environment characteristics. We partition tasks by file count, signal-to-noise ratio (SNR = #key files / #total files), and total data volume (i.e., file size in GB), then measure execution accuracy of GPT-5.5 with OpenHands within each partition. The results are shown in Figure 4.

**Data Complexity Degrades Performance.** We observe that increased environment complexity consistently impairs agent performance. File count shows a negative correlation with accuracy, though with substantial variance across communities. More notably, signal-to-noise ratio emerges as a strong predictor of difficulty. Communities with low SNR predominantly achieve lower accuracy, while those with high SNR perform considerably better. This pattern indicates that agents struggle primarily with distinguishing relevant data from semantically similar distractors rather than with navigating large file counts. Our community-based construction, which uses related Kaggle datasets, creates semantic ambiguity that misleads agent exploration. These findings validate that CODA-BENCH poses data intelligence challenges through environmental complexity.

**Large Data Volumes Create Bottlenecks.** We find that total data volume demonstrates a pronounced negative correlation with performance. Communities with data volumes below 3GB maintain relatively stable accuracy, while those exceeding this threshold exhibit consistent performance degradation. Several communities with volumes above 8GB drop to near-zero accuracy. This pattern indicates that large-scale data reading and exploration create substantial bottlenecks for current agents, likely due to the difficulty of processing large files during exploratory analysis. These findings highlight the limitations of current agents in handling production environments with 10GB-scale datasets.

### 6.3. Analysis of Interaction Behavior

To understand the relationship between agent behavior and task success, we analyze how the number of interaction rounds correlates with performance. We measure the average interaction rounds and execution accuracy for each agent. Figure 6 presents the results.

**Agent Frameworks Shape Interaction Efficiency.** We observe large variation in the number of interaction rounds

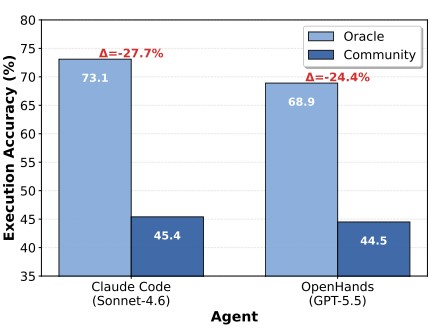

*Figure 5.* Results under giving oracle data or discovering in community.

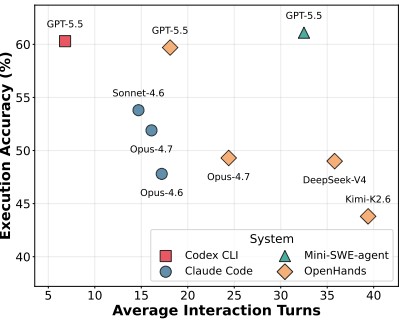

*Figure 6.* Results over various interaction rounds of agents.

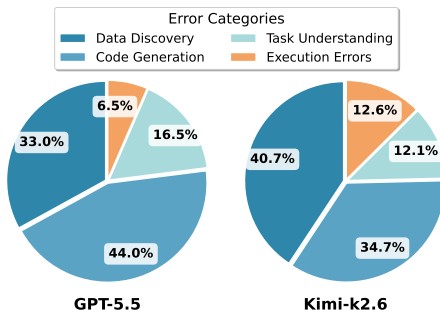

*Figure 7.* Error attribution of agents across failed cases.

across agent frameworks, even when they use the same underlying model. For GPT-5.5, Codex CLI achieves 60.3% EA in 6.8 rounds on average, compared with 61.1% EA in 32.5 rounds for Mini-SWE-agent and 59.7% EA in 18.1 rounds for OpenHands. Despite comparable execution accuracy, the required number of rounds differs by up to nearly 5×. This pattern holds for Claude models as well: Claude Code demonstrates superior efficiency with Sonnet-4.6 (14.7 rounds, 53.8% EA) and Opus-4.7 (16.1 rounds, 51.9% EA). These results indicate that native CLI tools achieve competitive accuracy with significantly fewer interactions through better optimization and tighter integration between models and execution environments.

**Model Capability Determines Performance Ceiling.** Within the same OpenHands framework, we observe that different models exhibit distinct interaction patterns. DeepSeek-V4-Pro requires 35.8 rounds to reach 49.0% EA, while Opus-4.7 achieves similar performance (49.3% EA) with only 24.4 rounds—DeepSeek requires 1.5× more interactions for equivalent results. More strikingly, Kimi-K2.6 requires even more rounds (39.4) yet achieves lower accuracy (43.8% EA), demonstrating that increased interaction alone cannot overcome fundamental model limitations. This ceiling effect reveals a fundamental bottleneck in data-intensive tasks, consistent with recent findings that uncertainty can accumulate across multi-step LLM-agent reasoning and that data quality issues can degrade machine learning performance (Zhao et al., 2025; Mohammed et al., 2025). Once an agent discovers incorrect data files during exploration, subsequent code modifications cannot recover from this error. No amount of debugging or code refinement compensates for operating on irrelevant data (Austin et al., 2021). This finding underscores that data discovery errors propagate irreversibly through the analytical pipeline, highlighting the critical importance of data intelligence in CODA-BENCH.

### 6.4. Error Attribution

To understand where agents fail, we manually analyzed 200 randomly sampled failure cases and categorized them into four error types based on the stage at which failure occurred: data discovery errors, task understanding errors, code generation errors, and execution errors. Specifically, data discovery errors refer to failures in locating the relevant files; task understanding errors occur when the agent misinterprets the question or data; code generation errors involve incorrect analysis logic; and execution errors arise from runtime failures such as crashes, timeouts, or dependency issues. We compare the error distributions between a high-performing agent (GPT-5.5) and a mid-tier agent (Kimi-K2.6), both using the OpenHands framework, to examine how failure modes vary across capability levels. The results are shown in Figure 7.

The two models exhibit different failure profiles. For GPT-5.5, code generation is the largest source of failure (44.0%), followed by data discovery (33.0%). For Kimi-K2.6, data discovery becomes the dominant failure type (40.7%), followed by code generation (34.7%). Kimi-K2.6 also has a higher proportion of execution errors than GPT-5.5 (12.6% vs. 6.5%). These results suggest that weaker models fail more often in identifying relevant data, whereas stronger models shift more of their errors toward analytical reasoning and code formulation. This capability-dependent shift supports the value of CODA-BENCH for jointly evaluating data discovery and code-based reasoning in realistic data science workflows.

## 7. Conclusion

In this paper, we introduce CODA-BENCH, the first benchmark designed to jointly evaluate the code and data intelligence of agents. Built upon the Kaggle ecosystem, CODA-BENCH leverages a large-scale data network to construct verifiable tasks and data-intensive environments. Evaluations on CODA-BENCH reveal that current agents still face significant challenges in solving complex problems under data-intensive settings, highlighting substantial room for improvement and providing a foundational benchmark for future research on integrated code and data intelligence.

## Acknowledgements

We thank all the anonymous reviewers for their insightful and valuable comments. This work was partially supported by the Scientific Research Innovation Capability Support Project for Young Faculty (Grant No. SRICSPYF-ZY2025001) and the National Natural Science Foundation of China (Grant Nos. 62436010, 62441230).

## Impact Statement

This paper presents work whose goal is to advance the field of Machine Learning. There are many potential societal consequences of our work, none of which we feel must be specifically highlighted here.

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

## A. Ethical Statement

We build CODA-BENCH based on datasets and notebooks sourced from Kaggle[7], a widely used data science platform. All datasets employed in this work are distributed under open licenses that allow academic research and redistribution, including Creative Commons licenses (CC BY, CC BY-SA, CC0) and Open Data Commons licenses (ODC-BY, PDDL). We carefully verify the licensing terms of each dataset to ensure full compliance with their respective usage requirements. We exclude personally identifiable information and sensitive data. The benchmark is intended solely for research.

## B. Graph Construction and Community Detection

### B.1. Graph Construction and Community Detection

**Co-occurrence Network Construction.** To systematically identify semantically related dataset clusters, we constructed a large-scale co-occurrence graph from all available Kaggle datasets and their associated notebooks. This graph captures real-world dataset usage patterns: when data scientists tackle similar problems, they tend to combine the same sets of datasets. The co-occurrence network is defined as follows:

- **Datasets (nodes)**: Each node represents a unique Kaggle dataset, which may contain one or more individual data files.
- **Data (i.e., files)**: The raw files (e.g., CSV, Excel, JSON, Parquet, images, and PDFs) contained within datasets, totaling 529,739 files across all datasets.
- **Co-occurrence edges**: An edge connects two datasets if they appear together in at least one Kaggle notebook, with edge weights indicating the number of such co-occurrences.

Table 5 summarizes the statistics of the whole graph.

**Community Detection.** We applied the Leiden algorithm (Traag et al., 2019) with resolution parameter $\gamma = 1.0$, which identified 323 communities with a modularity score of 0.711. This high modularity indicates strong community structure—datasets within the same community co-occur far more frequently than those from different communities, validating that our graph effectively captures thematic coherence among datasets.

Figure 3 and Figure 11 visualize this network with community assignments, where each node represents a dataset and nodes of the same color belong to the same community detected by the Leiden algorithm. The spatial layout reveals the inherent structure of the data science ecosystem: densely connected clusters correspond to thematically coherent dataset groups, while bridge nodes connect different domains. To facilitate future research and enable interactive exploration of this dataset ecosystem, we will release an online demo of the complete network visualization.

*Table 5.* Statistics of co-occurrence graph used in CODA-BENCH.

| Metric | All Kaggle | Selected in CODA-BENCH |
|---|---|---|
| Datasets (nodes) | 21,122 | 829 |
| Data (i.e., avg files) | 25.08 | 29.18 |
| Co-occurrences (edges) | 93,727 | 1,903 |
| Average degree | 8.87 | 4.59 |
| Maximum degree | 412 | 39 |
| Graph density | 0.00042 | 0.00179 |
| Clustering coefficient | 0.564 | 0.0822 |
| Communities detected | 323 | 31 |
| Modularity | 0.711 | – |

**Benchmark Curation.** From the 323 detected communities, we carefully selected 31 communities (829 datasets) that are highly relevant to practical data analysis tasks. This selection ensures CODA-BENCH covers diverse, real-world data science scenarios while maintaining coherent thematic groupings within each community. Table 5 summarizes the statistics at different levels of our graph construction pipeline, and Figure 11 shows the filtered network structure.

### B.2. Community Analysis

Table 6 presents 10 sampled communities, revealing the breadth of CODA-BENCH's coverage across data science domains, from foundational ML benchmarks and healthcare analytics to entertainment recommendation systems and geospatial pandemic analysis. Some example community themes include:

- **community_0** (154 datasets): Foundational machine learning benchmarks including Iris Species, Pima Indians Diabetes, and Credit Card Fraud Detection. These datasets are frequently used for teaching and basic ML experiments.

---

[7] https://www.kaggle.com

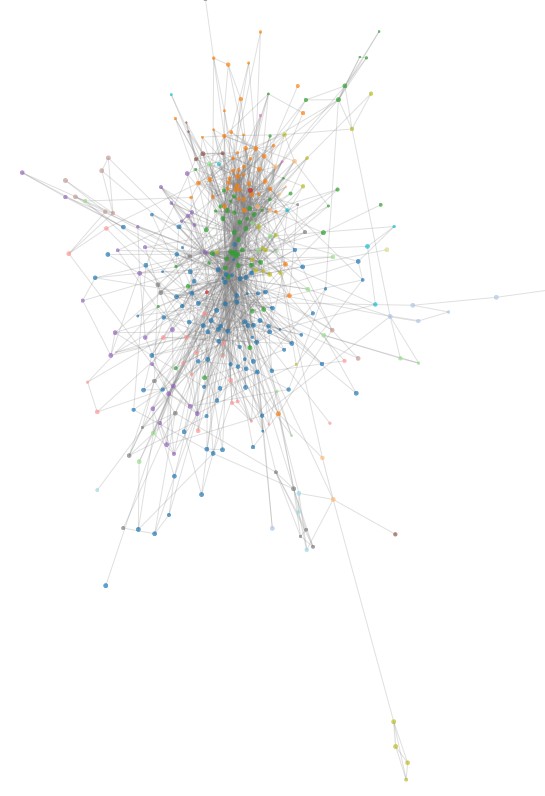

*Figure 8.* Network of 31 selected communities used in CODA-BENCH. Node colors indicate community membership, node sizes reflect notebook usage frequency, and edge widths represent co-occurrence strength. The network exhibits clear clustering patterns corresponding to different data science domains.

*Table 6.* 10 sampled communities in CODA-BENCH.

| Rank | Community ID | Datasets | Notebooks | Dominant Theme |
|------|--------------|----------|-----------|----------------|
| 1 | community_0 | 154 | 7,868 | Classic ML benchmarks (Iris, Diabetes, Credit Card Fraud) |
| 2 | community_2 | 88 | 2,778 | COVID-19 & global geography/demographics |
| 3 | community_4 | 70 | 3,295 | Popular mixed datasets (COVID-19, Netflix, Airbnb, Olympics) |
| 4 | community_8 | 47 | 2,099 | Entertainment & media (MovieLens, Netflix, Video Games, TMDB) |
| 5 | community_27 | 32 | 1,298 | Health & lifestyle (Smoker, Wine Quality, Iris, Titanic) |
| 6 | community_15 | 28 | 1,063 | India-specific data (COVID-19, Air Quality, Unemployment) |
| 7 | community_33 | 28 | 974 | Finance & credit (Yelp, Lending Club, Stock, Credit Risk) |
| 8 | community_24 | 24 | 622 | Meta-Kaggle (ML Surveys, arXiv, Competition Data) |
| 9 | community_25 | 19 | 719 | Recommendation systems (MovieLens 20M, Online Retail) |
| 10 | community_90 | 15 | 859 | Healthcare prediction (Depression, Horse Survival, Obesity) |
| **Total (all 31 communities)** | | 829 | 30,624 | |

- **community_2** (88 datasets): COVID-19 pandemic analysis datasets combined with geographical, demographic, and country-level statistics. Reflects the surge in COVID-19 data science during 2020-2021.
- **community_4** (70 datasets): Diverse popular datasets spanning multiple domains including pandemic data, entertainment (Netflix, Airbnb), and sports (Olympics). These datasets are frequently used in exploratory data analysis tutorials.
- **community_8** (47 datasets): Entertainment and media analytics including movie recommendations (MovieLens), streaming platforms (Netflix), video games, and movie databases (TMDB).
- **community_27** (32 datasets): Mixed health and lifestyle datasets including smoking prediction, wine quality assessment, and classic benchmarks like Iris and Titanic.

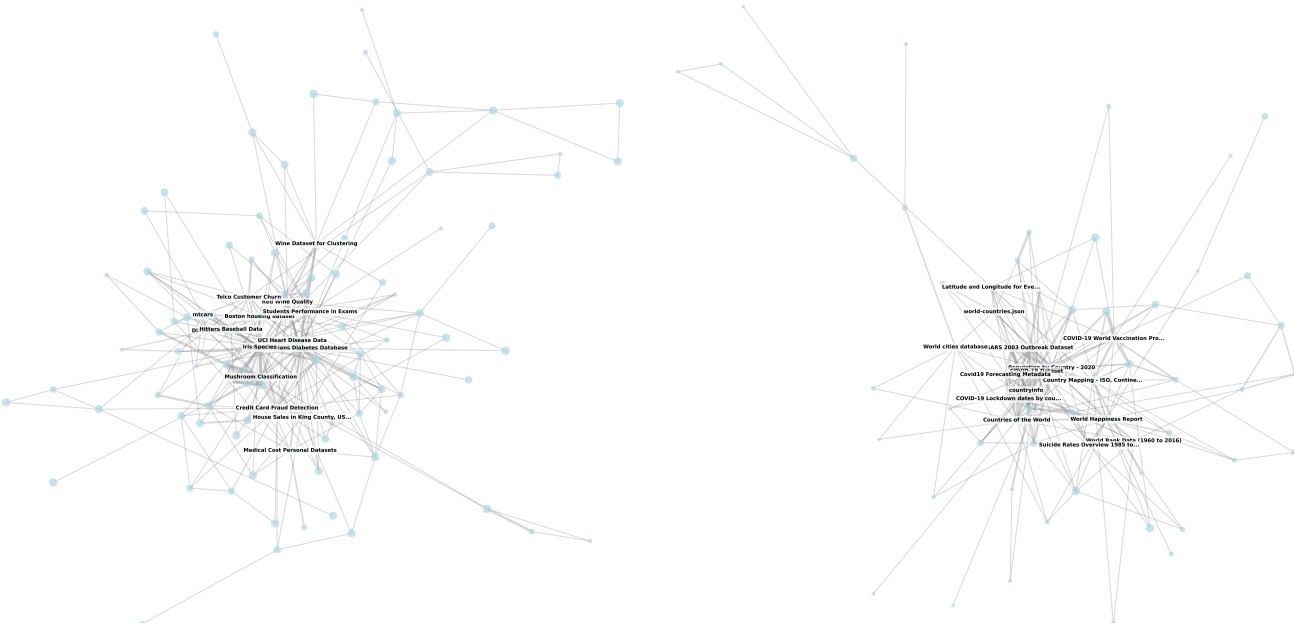

*Figure 9.* Detailed visualization of community_0 (Classic ML Benchmarks). The network shows 130 connected datasets (after removing 24 isolated nodes). Node sizes represent notebook usage frequency. Labels indicate the 15 most central datasets by degree. Key datasets include Iris Species, Pima Indians Diabetes, Credit Card Fraud Detection, Water Quality, and Medical Insurance.

*Figure 10.* Detailed visualization of community_2 (COVID-19 & Global Geography). The network shows 70 connected datasets (after removing 18 isolated nodes). Labels highlight the 15 most central datasets. The community reflects the integration of pandemic data with country-level statistics for comparative analysis.

**Community 0: Classic ML Benchmarks** (154 datasets, Figure 9) forms the largest and most interconnected community, anchored by foundational datasets that have shaped machine learning education and research. Central datasets include Iris Species (the iconic classification benchmark), Pima Indians Diabetes Database, Credit Card Fraud Detection (a challenging imbalanced classification problem), Water Quality, Medical Insurance, and Telco Customer Churn. The dense connectivity within this community reflects how practitioners frequently combine these datasets for comparative analysis and pedagogical purposes.

**Community 2: COVID-19 & Global Geography** (88 datasets, Figure 10) exemplifies how real-world events drive dataset ecosystem evolution. This community emerged from the unprecedented surge in pandemic-related data science during 2020–2021, integrating COVID-19 statistics with geographical, demographic, and socioeconomic indicators for comparative country-level analysis. Key datasets include:

- **countryinfo**: Country-level metadata (98 notebooks, 3,870 downloads)
- **COVID-19 Dataset**: Core pandemic statistics (97 notebooks, 400,604 downloads)
- **Daily Temperature of Major Cities**: Climate data (97 notebooks, 45,957 downloads)
- **Countries of the World**: Country profiles (96 notebooks, 67,909 downloads)
- **Suicide Rates Overview 1985–2016**: Mental health statistics (96 notebooks, 148,681 downloads)
- **World Happiness Report**: Country happiness scores (95 notebooks, 369,267 downloads)
- **COVID-19 World Vaccination Progress**: Vaccination tracking (95 notebooks, 113,598 downloads)

Table 7 provides a complete list of all 88 datasets in community 2, sorted by their degree (number of co-occurrence connections).

*Table 7.* Complete list of datasets in community_2 (COVID-19 & Global Geography), sorted by degree.

| # | Title | Dataset Slug | Degree | Downloads |
|---|---|---|---|---|
| 1 | COVID-19 Dataset | imdevskp/corona-virus-report | 166 | 400,604 |
| 2 | World Happiness Report | unsdsn/world-happiness | 134 | 369,267 |
| 3 | Population by Country - 2020 | tanuprabhu/population-by-country-2020 | 114 | 26,927 |
| 4 | Country Mapping - ISO, Continent, Region | andradaolteanu/country-mapping-iso-continen | 94 | 17,476 |
| 5 | countryinfo | koryto/countryinfo | 89 | 3,870 |
| 6 | Countries of the World | fernandol/countries-of-the-world | 83 | 67,909 |
| 7 | Latitude and Longitude for Every Country | paultimothymooney/latitude-and-longitude-f | 66 | 18,576 |
| 8 | COVID-19 World Vaccination Progress | gpreda/covid-world-vaccination-progress | 61 | 113,598 |
| 9 | Suicide Rates Overview 1985 to 2016 | russellyates88/suicide-rates-overview-1985 | 48 | 148,681 |
| 10 | World cities database | juanmah/world-cities | 47 | 20,577 |
| 11 | SARS 2003 Outbreak Dataset | imdevskp/sars-outbreak-2003-complete-datas | 45 | 8,846 |
| 12 | Covid19 Forecasting Metadata | rohanrao/covid19-forecasting-metadata | 43 | 1,646 |
| 13 | Health Nutrition and Population Stat. | theworldbank/health-nutrition-and-populati | 38 | 20,169 |
| 14 | world-countries.json | ktochylin/world-countries | 35 | 4,782 |
| 15 | COVID-19 Lockdown dates by country | jcyzag/covid19-lockdown-dates-by-country | 33 | 4,715 |
| 16 | country to continent | statchaitya/country-to-continent | 32 | 8,363 |
| 17 | 2020 Cost of Living | andradaolteanu/2020-cost-of-living | 31 | 1,431 |
| 18 | SmokingStats | osciiart/smokingstats | 31 | 543 |
| 19 | Life Expectancy (WHO) | kumarajarshi/life-expectancy-who | 28 | 177,422 |
| 20 | Covid-19 Global Dataset | josephassaker/covid19-global-dataset | 25 | 18,397 |
| 21 | World Bank Data (1960 to 2016) | gemartin/world-bank-data-1960-to-2016 | 25 | 6,931 |
| 22 | World Bank WDI 2.12 - Health Systems | danevans/world-bank-wdi-212-health-systems | 25 | 6,090 |
| 23 | US Accidents (2016 - 2023) | sobhanmoosavi/us-accidents | 23 | 164,536 |
| 24 | World Population 1960-2018 | imdevskp/world-population-19602018 | 23 | 6,718 |
| 25 | Daily Temperature of Major Cities | sudalairajkumar/daily-temperature-of-major | 22 | 45,957 |
| 26 | World Happiness Report 2020 | londeen/world-happiness-report-2020 | 21 | 5,430 |
| 27 | 2019 Coronavirus dataset (Jan-Feb 2020) | brendaso/2019-coronavirus-dataset-01212020 | 20 | 17,995 |
| 28 | China Regions Map | gpreda/china-regions-map | 20 | 2,018 |
| 29 | CO2 Emissions | ulrikthygepedersen/co2-emissions-by-countr | 19 | 8,600 |
| 30 | Python Folium Country Boundaries | subota/python-folio-country-boundaries | 19 | 404 |
| 31 | COVID19 Global Weather Data | winterpierre91/covid19-global-weather-data | 17 | 1,619 |
| 32 | COVID-19 Tracking Germany | headsortails/covid19-tracking-germany | 17 | 9,101 |
| 33 | Human Development Reports | sudhirnl7/human-development-index-hdi | 17 | 2,201 |
| 34 | World Happiness Report 2023 | ajaypalsinghlo/world-happiness-report-2023 | 16 | 11,896 |
| 35 | ASHRAE Global Thermal Comfort Database | claytonmiller/ashrae-global-thermal-comfor | 15 | 3,074 |
| 36 | Automobile Dataset | toramky/automobile-dataset | 15 | 80,093 |
| 37 | Human Development World Index | iamsouravbanerjee/human-development-index- | 15 | 4,587 |
| 38 | Countries ISO Codes — Continent — Flags | andreshg/countries-iso-codes-continent-fla | 12 | 1,812 |
| 39 | COVID19 Worldwide Testing Data | lin0li/covid19testing | 12 | 4,175 |
| 40 | Global Food Prices | jboysen/global-food-prices | 12 | 15,639 |
| 41 | Paris 2024 Olympics Medals | berkayalan/paris-2024-olympics-medals | 12 | 9,850 |
| 42 | Temperature change | sevgisarac/temperature-change | 12 | 31,182 |
| 43 | COVID-19 data from John Hopkins Univ. | antgoldbloom/covid19-data-from-john-hopkin | 11 | 23,706 |
| 44 | Corporate Environmental Impact | mannmann2/corporate-environmental-impact | 10 | 1,843 |
| 45 | HR Analytics | giripujar/hr-analytics | 10 | 33,783 |
| 46 | Who eats the food we grow? | dorbicycle/world-foodfeed-production | 9 | 17,874 |
| 47 | econfin | zhaofengchen/econfin | 8 | 59 |
| 48 | GDP World Bank Data | ibrahimmukherjee/gdp-world-bank-data | 8 | 2,840 |
| 49 | Mental Health and Suicide Rates | twinkle0705/mental-health-and-suicide-rate | 8 | 17,656 |
| 50 | UP School Women in Datathon — Dataset | upschoolio/up-school-women-in-datathon-dat | 8 | 153 |
| 51 | Country Coordinates GeoJson | danielvalyano/country-coord | 7 | 407 |
| 52 | Income by Country | frankmollard/income-by-country | 7 | 3,686 |
| 53 | Olympic Summer & Winter Games, 1896-2022 | piterfm/olympic-games-medals-19862018 | 7 | 12,855 |
| 54 | Tokyo 2020 Olympics Medals | berkayalan/2021-olympics-medals-in-tokyo | 6 | 6,454 |
| 55 | Germany COVID-19 (jan-September) | akshat0007/germany-covid19-janseptember | 6 | 210 |
| 56 | Haberman's Survival Data Set | gilsousa/habermans-survival-data-set | 6 | 38,332 |
| 57 | Household Electric Power Consumption | uciml/electric-power-consumption-data-set | 5 | 58,323 |
| 58 | Global Child Mortality Rate | drateendrajha/global-child-mortality-rate | 5 | 868 |
| 59 | Global Terrorism Report for World Happ. | berkantaslan/global-terrorism-report-for-w | 5 | 57 |
| 60 | world happiness report 2022 | ajaypalsinghlo/world-happiness-report-2022 | 5 | 8,266 |
| 61 | Yearly Air Quality Index (AQI) for CDP | reubencpereira/yearly-air-quality-index-aq | 5 | 200 |
| 62 | Countries Population | centurion1986/countries-population | 4 | 667 |
| 63 | Countries Travel inbound dataset (1995-2018) | namanphy7/countries-travel-inbound-dataset | 4 | 230 |
| 64 | lateset-covid | zhaofengchen/latesetcovid | 4 | 44 |
| 65 | Opinion Lexicon English | rafay12/opinion-lexicon-english | 4 | 106 |
| 66 | us states map | satyabrataroy/us-states-map | 4 | 583 |
| 67 | data_measure | flyingsolo/data-measure | 3 | 18 |
| 68 | Global Commodity Trade Statistics | unitednations/global-commodity-trade-stati | 3 | 13,556 |
| 69 | india-climate | flyingsolo/indiaclimate | 3 | 27 |
| 70 | municipiosbrasileiros | educfrio/municipiosbrasileiros | 3 | 377 |
| 71 | olimpiadas | luciotinnirellohsbc/olimpiadas | 3 | 50 |

*Continued on next page*

Table 7 Continued from previous page

| # | Title | Dataset Slug | Degree | Downloads |
|---|-------|-------------|--------|-----------|
| 72 | trip_advisor_data | mintylife/trip-advisor-data | 3 | 35 |
| 73 | 30 Years of European Wind Generation | sohier/30-years-of-european-wind-generatio | 2 | 2,791 |
| 74 | COVID-19 in Poland Dataset | fischerbach/covid19-in-poland-dataset | 2 | 323 |
| 75 | Geolocation Data [Longitude Latitude] | liewyousheng/geolocation | 2 | 3,856 |
| 76 | interventions | ilkeakar/interventions | 2 | 19 |
| 77 | Italian Regions | ludovicoristori/italian-regions | 2 | 231 |
| 78 | new_data_additions | sunnyfunny/new-data-additions | 2 | 23 |
| 79 | SIIM-FISABIO-RSNA Covid 2021 | andradaolteanu/siimfisabiorsna-covid-2021 | 2 | 62 |
| 80 | Statistics of Summer Olympics- Tokyo 2020 | hamdallak/statistics-of-summer-olympics-to | 2 | 391 |
| 81 | ASHRAE thermal comfort dataset | khorikoshi/ashrae-thermal-comfort-dataset | 1 | 50 |
| 82 | Charity Navigator Scores Expenses Dataset | katyjqian/charity-navigator-scores-expense | 1 | 1,271 |
| 83 | DIVI Intensivregister | bboyhusky/divi-intensivregister | 1 | 59 |
| 84 | final_cars_data | jakubmalachowski/final-cars-data | 1 | 12 |
| 85 | World History of Wars and Demographics | mattiaperozzi/history-of-demographics-and- | 1 | 365 |
| 86 | images-ann-ibm | rajmehra03/imagesannibm | 1 | 32 |
| 87 | iraq_cities | linhvuu/iraq-cities | 1 | 8 |
| 88 | WHO Physical Activity-Country Profile 2022 | yingwoowang/who-physical-activity-country- | 1 | 80 |

**Total: 88 datasets**

## C. Example of Benchmark Construction

We illustrate the construction of tasks in CODA-BENCH through a complete example, demonstrating how an initial task derived from a Kaggle notebook solution progressively evolves into a challenging yet solvable benchmark task.

**Model Pool for Adversarial Evolution.** Our framework employs an ensemble of four advanced LLMs as the discriminator pool: GPT-5.2 (OpenAI), Claude-Sonnet-4.5 (Anthropic), Gemini-3.0-Flash-Preview (Google), and Kimi-K2 (Moonshot AI). At each iteration, we randomly sample 3 models as discriminators to solve the task, while the remaining model serves as the generator to propose evolution strategies. This rotation mechanism ensures that evolved tasks are not tailored to exploit weaknesses of any single model, but rather capture genuine analytical challenges that generalize across diverse model architectures.

**Evolution Process Overview.** The complete construction pipeline proceeds through five steps:

1. Solution Anchor Identification: Extract verifiable numerical results from Kaggle notebook cells as ground-truth anchors
2. Initial Question Generation: Formulate natural language questions from solution anchors using LLM
3. Iterative Evolution: Apply adversarial refinement to increase task difficulty while preserving answer uniqueness
4. Difficulty Validation: Measure solve rate degradation across iterations to confirm increased challenge
5. Human Verification: Final quality check ensuring tasks meet all criteria (unambiguous, self-contained, verifiable, non-trivial, authentic)

**Pseudocode and Workflow for Adversarial Evolution**

Algorithm 1 formalizes the adversarial evolution procedure described in the main text.

---

**Algorithm 1** Adversarial Task Evolution

---

**Require:** Verified solution anchor $A$, discriminator models $\mathcal{D}$, generator model $G$
**Require:** Difficulty threshold $\tau = 0.667$, max iterations $T = 5$
**Ensure:** Evolved task $Q$ or rejection
1: $Q_0 \leftarrow G.\text{GenerateInitialQuestion}(A)$
2: **for** $t = 1$ to $T$ **do**
3:     Sample 3 models $D_1, D_2, D_3 \sim \mathcal{D}$
4:     solve_rate $\leftarrow \frac{1}{3}\sum_{i=1}^{3} \mathbb{K}[D_i \text{ solves } Q_{t-1}]$
5:     **if** solve_rate $> \tau$ **then**
6:         $Q_t \leftarrow G.\text{IncreaseDifficulty}(Q_{t-1})$
7:     **else if** solve_rate $= 0$ **then**
8:         diagnosis $\leftarrow G.\text{DiagnoseFailure}(Q_{t-1}, \{D_i\})$
9:         **if** diagnosis $=$ TYPE_1_DEFECT **then**
10:             $Q_t \leftarrow G.\text{RepairTask}(Q_{t-1})$
11:         **else if** diagnosis $=$ TYPE_3_AMBIGUOUS **then**
12:             $Q_t \leftarrow G.\text{RefineAmbiguity}(Q_{t-1})$
13:         **else**
14:             **return** REJECT (genuine difficulty)
15:         **end if**
16:     **else**
17:         Verify $Q_{t-1}$ with human annotator
18:         **if** verified **then**
19:             **return** $Q_{t-1}$
20:         **else**
21:             **return** REJECT
22:         **end if**
23:     **end if**
24: **end for**
25: **return** REJECT (non-convergence)

---

**Step 1: Solution Anchor Identification**

---

**Source: Kaggle Notebook Solution**

**Notebook:** netflix-data-analysis (Cell 15)

**Code:**

```python
# Obtain the content category count
category_dict = {}
for category in netflix_contents.query("type == 'TV Show'").listed_in.dropna():
    category_split = category.split(", ")
    for splited_category in category_split:
        if splited_category not in category_dict:
            category_dict[splited_category] = 1
        else:
            category_dict[splited_category] += 1

# Get sorted lists of category frequency and category names
category_frequencies, category_names = zip(*sorted(
    zip(category_dict.values(), category_dict.keys()), reverse=True))

category_frequency_df = pd.DataFrame({
    "category": category_names,
    "frequency": category_frequencies
})
category_frequency_df["percentage"] = \
    round(category_frequency_df["frequency"] /
          category_frequency_df["frequency"].sum() * 100, 3)
```

**Markdown Output:**

"For TV shows, the top 5 frequently mentioned categories are **International TV Shows (1199)**, **TV Dramas (704)**, **TV Comedies (525)**, **Crime TV Shows (427)**, and **Kids' TV (414)**."

**Solution Anchor:** International TV Shows, 1199; TV Dramas, 704; TV Comedies, 525; Crime TV Shows, 427; Kids' TV, 414

**Verification:** Re-execution confirms deterministic reproducibility ($\epsilon < 10^{-6}$).

---

**Step 2: Initial Question Generation (Iteration 0)**

---

## V0: Initial Question (LLM Generated)

**Generated Question:**

"In the analysis of TV Show content categories, after splitting the 'listed_in' column to separate multiple categories per title, what are the top 5 most frequently listed categories and their respective title counts?"

**Discriminator Evaluation (3 models randomly sampled):**

- GPT-5.2: ✓Correct
- Claude-Sonnet-4.5: ✓Correct
- Gemini-3.0-Flash-Preview: ✓Correct

**Generator:** Kimi-K2 (proposes evolution strategy)

**Solve Rate:** $r^{(0)} = 3/3 = 100\% \rightarrow$ Task too easy, trigger evolution.

**Generator Analysis:**

```
# Inspect successful trajectories
$ analyze_solutions(discriminator_outputs)
```

**Identified Issues:**

1. **Verbose preamble:** "In the analysis of TV Show content categories" provides unnecessary context
2. **Explicit hint:** "after splitting the 'listed_in' column" directly reveals the solution approach

---

**Step 3: First Evolution Iteration**

---

## V1: First Iteration (Remove Redundancy)

**Evolved Question (V1):**

"What are the top 5 most frequently listed categories for TV shows and their respective title counts?"

**Evolution Strategy:**

```
{
  "prefix_removed": [
    "In the analysis of TV show content categories,"
  ],
  "inferable_info_removed": [
    "after splitting entries that contain multiple categories"
  ]
}
```

**Human Review:** ✓Approved (unambiguous, no information loss)

**Discriminator Re-evaluation (same 3 models):**

- GPT-5.2: ✓Correct
- Claude-Sonnet-4.5: ✓Correct
- Gemini-3.0-Flash-Preview: ✓Correct

**Generator:** Kimi-K2 (proposes next evolution)

**Solve Rate:** $r^{(1)} = 3/3 = 100\% \rightarrow$ Continue evolution.

**Generator Analysis:**

1. **Column name hint:** "listed categories" closely resembles column name 'listed_in'
2. **Domain hint:** "title counts" explicitly reveals the dataset domain (media titles)

**Step 4: Second Evolution Iteration**

---

### V2: Second Iteration (Generalize Terms)

**Evolved Question (V2):**

"What are the top 5 most frequently assigned genres for TV shows and their counts?"

**Evolution Strategy:**

```
{
  "simplifications_made": [
    {
      "type": "column_name_generalized",
      "original": "listed categories",
      "simplified": "assigned genres",
      "reasoning": "Term 'listed categories' mirrors column name 'listed_in'. Using '
          genres' is more generic – a data analyst would naturally map it to the
          appropriate column without direct hint."
    },
    {
      "type": "hint_removed",
      "original": "respective title counts",
      "simplified": "their counts",
      "reasoning": "Word 'title' explicitly hints at media domain. Simply 'counts'
          maintains frequency requirement without domain revelation."
    }
  ]
}
```

**Discriminator Re-evaluation (3 new models sampled):**
- Kimi-K2: ✓Correct
- Claude-Sonnet-4.5: ✓Correct
- Gemini-3.0-Flash-Preview: ✗Incorrect

**Generator:** GPT-5.2 (evaluates evolution potential)

**Solve Rate:** $r^{(2)} = 2/3 = 66.7\% \rightarrow$ **Difficulty increased as intended**

**Generator Decision:**

```
$ check_evolution_potential(v2_question)
# Output: "No further generalizations possible without introducing ambiguity"
# Reason: 'genres' is already maximally generic; removing 'TV shows' would
#         make question ambiguous (Movies vs. TV Shows)
```

**Termination Condition Met:** Task remains solvable but cannot be further simplified without ambiguity.

**Step 5: Human Verification and Finalization**

---

### Final Task (Human Verification)

**Final Task:**

"What are the top 5 most frequently assigned genres for TV shows and their counts?"

**Expected Answer:** International TV Shows, 1199; TV Dramas, 704; TV Comedies, 525; Crime TV Shows, 427; Kids' TV, 414

**Human Verification Checklist:**
- ✓ **Unambiguous:** Question has single valid interpretation
- ✓ **Self-contained:** No missing information required to solve
- ✓ **Verifiable:** Ground-truth answer is deterministically reproducible
- ✓ **Non-trivial:** Requires multi-step reasoning (filter TV shows → split genres → aggregate → rank)
- ✓ **Authentic:** Reflects real data analysis workflow from Kaggle notebook

**Status: Accepted into CODA-BENCH** (Evolution iterations: 2, Total evaluations: 9)

---

**Evolution Trajectory Analysis.** Through this example, we demonstrate how solve rate decreases from 100% (V0) to 66.7% (V2). As shown in Table 8, there is a clear trade-off between task conciseness and difficulty. Notably, the most effective evolution (V1→V2) achieved difficulty increase not by adding complexity, but by *removing* domain-specific hints, generalizing "movie" to "content" forced models to infer context from the dataset rather than relying on lexical cues.

*Table 8.* Evolution trajectory of task.

| Version | Length | Key Change | Solve Rate |
|---------|--------|-----------|-----------|
| V0 | 38 words | Initial generation | 100% |
| V1 | 16 words | Remove verbose preamble | 100% |
| V2 | 14 words | Generalize terminology | 66.7% |

**Quality Control through Human Verification.** Not all evolution attempts succeed. Some introduced harmful ambiguity (e.g., "What are the top genres?" without specifying count or ranking criteria), which models correctly identified as task defects rather than genuine challenges. Our framework's diagnostic analysis automatically detected such cases and reverted to previous versions. Additionally, human reviewers rejected 12% of evolved tasks due to over-generalization, ensuring that increased difficulty stems from analytical complexity rather than specification flaws.

## D. Tasks Illustration

Here, we present a complete task example from CODA-BENCH, illustrating the full specification of a benchmark instance. Each task provides models with: (1) a natural language question describing the analytical objective, (2) a data environment containing intensive data. Agents are required to autonomously explore the provided data environment, determine the appropriate analytical approach, implement the solution in code, and produce the final answer. To enable rigorous evaluation, we provide reference answers along with reference solutions that ensure consistent and reproducible assessment across different model outputs.

## Input

### Question

How many rows represent cancellations and what percentage of all rows does this represent?

### Environment

```
/data/community_0/
+-- 2000-16-traffic-flow-england-scotland-wales/
|   +-- source/
|      +-- Areas.shp
|      +-- accidents_2005_to_2007.csv
|      +-- ukTrafficAADF.csv
|      +-- ...
+-- onlineretail/
|   +-- source/
|      +-- OnlineRetail.csv
+-- credit-card/
|   +-- source/
|      +-- application_data.csv
|      +-- columns_description.csv
|      +-- previous_application.csv
+-- fraud-detection/
|   +-- source/
|      +-- fraudTest.csv
|      +-- fraudTrain.csv
+-- ecommerce-data/
|   +-- source/
|      +-- data.csv
+-- ...  (441 total files across 154 datasets)
```

File path: onlineretail/source/OnlineRetail.csv
File size: 45.6 MB
Number of rows: 541,909
Number of columns: 8
Relevant columns: Quantity

Signal-to-noise ratio: 0.002
Distractor similarity: High

## Output

### Ground Truth Answer

*10624; 1.96%*

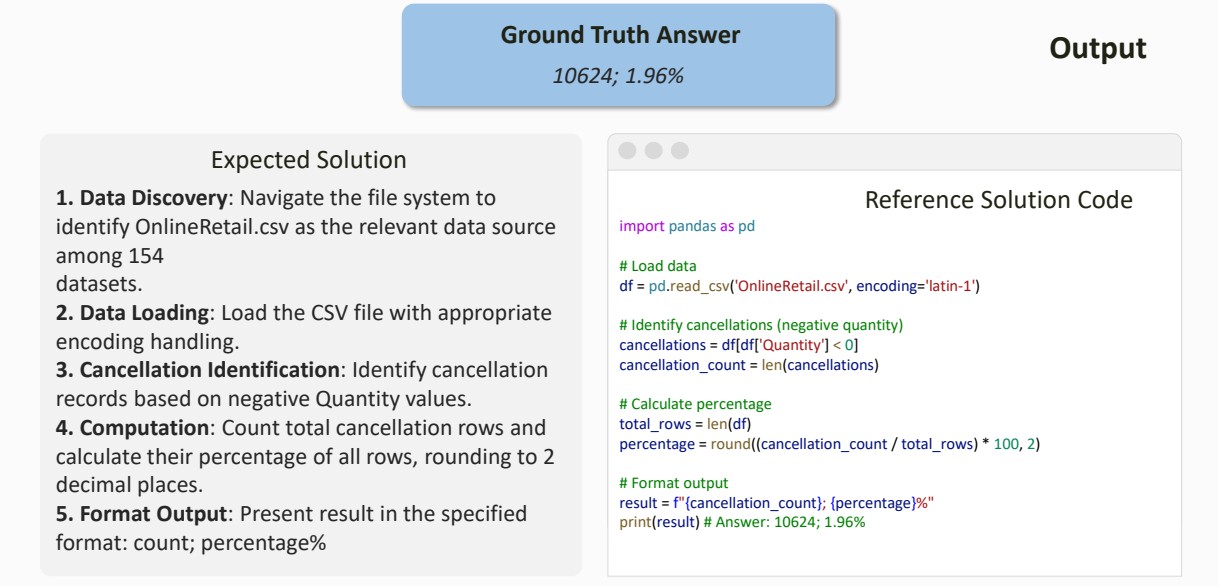

#### Expected Solution

**1. Data Discovery**: Navigate the file system to identify OnlineRetail.csv as the relevant data source among 154 datasets.
**2. Data Loading**: Load the CSV file with appropriate encoding handling.
**3. Cancellation Identification**: Identify cancellation records based on negative Quantity values.
**4. Computation**: Count total cancellation rows and calculate their percentage of all rows, rounding to 2 decimal places.
**5. Format Output**: Present result in the specified format: count; percentage%

#### Reference Solution Code

```python
import pandas as pd

# Load data
df = pd.read_csv('OnlineRetail.csv', encoding='latin-1')

# Identify cancellations (negative quantity)
cancellations = df[df['Quantity'] < 0]
cancellation_count = len(cancellations)

# Calculate percentage
total_rows = len(df)
percentage = round((cancellation_count / total_rows) * 100, 2)

# Format output
result = f"{cancellation_count}; {percentage}%"
print(result) # Answer: 10624; 1.96%
```

*Figure 11.* A representative task from our benchmark with all components

# E. Evaluation Sandbox

Here, we give the sandbox setup and evaluation protocols for CODA-BENCH. A key contribution of our benchmark is the provision of a reproducible, production-grade evaluation environment that closely mirrors real-world development scenarios while maintaining strict experimental control.

## E.1. Evaluation Environment

To ensure fair comparison and reproducibility, we developed a Docker-based sandbox infrastructure that provides isolated, standardized environments for each evaluation run. This design eliminates confounding factors from system-level variations and enables precise measurement of agent capabilities.

**Core Dependencies.** Each container is provisioned with Python 3.11 and a carefully curated set of data science libraries (pandas 2.1.0+, numpy 1.24.0+, matplotlib 3.7.0+, scikit-learn 1.3.0+), along with file format support (openpyxl for Excel, pyarrow for Parquet) and essential system tools (ls, grep, vim, curl, tree). This configuration reflects a realistic data analysis workspace, and agents are permitted to install any additional packages they require within the environment.

**Resource Constraints.** We impose practical resource limits, 4GB memory, 2 CPU cores, and a 600-second timeout per task, to simulate real-world computational constraints and prevent runaway executions. Crucially, data directories are mounted as read-only to enforce non-destructive analysis and prevent agents from circumventing task requirements through data modification.

**Directory Structure.** The workspace follows a standardized layout designed for clarity and ease of evaluation:

```
/workspace/
|-- task_description.txt       # Task instruction
|-- data/                       # Data environment
|   |-- dataset_1/
|   |-- dataset_2/
|   |-- ...
|-- result.txt                  # Agent output
```

## E.2. Agent Configurations

We evaluate both commercial CLI tools and open-source agent frameworks to provide comprehensive coverage of the current landscape.

**Native CLI Tools.** We benchmark two state-of-the-art commercial CLI tools under their default configurations: Claude Code (v2.1.150), and Codex CLI (v2.3.1). All tools utilize default temperature settings and their built-in code execution capabilities, ensuring that our evaluation reflects out-of-the-box performance without task-specific tuning.

**Open-source Agent Framework.** For framework-based evaluation, we employ OpenHands v1.7 and mini-swe-agent v2.0.0 with the temperature set to 0.

# F. Case Study

In this section, we present success and failure cases on our CODA-BENCH using GPT-5.5 and the OpenHands framework. GPT-5.5 (OpenHands) demonstrates both successful analytical pipelines and critical failure modes through multi-turn interactions.

**Success Case 1: Correct Dataset Discovery and Analysis**

---

### Success Case

**Task:** After sorting records by income (primary) and GDP per capita (secondary) in descending order, which country ranks first and what is its income?

**Expected Answer:** Qatar; 125,000.0    **Agent Answer:** Qatar; 125,000.0    ✓**CORRECT**

**Key Actions (20 messages):**

**Action 1-2:** ✓ Environment exploration

```
$ view task_description.txt
$ ls -1
# Output: 170+ datasets including "unsupervised-learning-on-country-data"
```

**Action 3:** ✓ Dataset discovery and inspection

```
$ python -c "import pandas as pd; df=pd.read_csv('unsupervised-learning-on-country-
    data/source/Country-data.csv'); print(df.shape, df.columns.tolist())"
# Output: (167, 10) ['country', 'child_mort', ..., 'income', ..., 'gdpp']
```

**Action 4:** ✓ Sorting verification

```
$ python -c "df.sort_values(['income','gdpp'], ascending=[False,False]).head(10)[['
    country','income','gdpp']]"
# Output: Qatar (125000, 70300) ranks first
```

**Action 5-7:** ✓ Create reproducible script and output

```
df = pd.read_csv("unsupervised-learning-on-country-data/source/Country-data.csv")
top = df.sort_values(["income", "gdpp"], ascending=[False, False]).iloc[0]
with open("result.txt", "w") as f:
    f.write(f"{top['country']}; {top['income']:,.1f}\n")
```

- - - - - - - - - - - - - - - - - - - - - - - - - - - - - - - - - - - - - - - - - - - - - - - - - - - - - - - - - - - - - - - - - - - - - - - -

**Success Factors:** Correct dataset identification → accurate data processing → proper output formatting.

---

**Failure Case 1: Dataset Discovery Error**

---

## Failure Case

**Task:** How many competitions enabled between May 2021 and June 2023 contain generative AI keywords ('deepfake', 'gpt', 'llm', 'GAN', etc.)?

**Expected Dataset:** meta_kaggle   **Agent Selected:** kaggles-all-completed-competition-dataset

**Expected Answer:** 10   **Agent Answer:** 2   ✗**INCORRECT**

**Key Actions (38 messages):**

**Action 1-2:** ✓ Environment exploration

```
$ view task_description.txt
$ ls -1
# Output: 26 datasets including 'meta_kaggle', 'kaggles-all-completed-competition-
    dataset'
```

**Action 3:** ✗ **Wrong dataset selection** (critical error)

```
$ python -c "df=pd.read_csv('kaggles-all-completed-competition-dataset/source/
    kaggle_comp_submission.csv'); print(df.shape, df['start_dt'].max())"
# Output: (527, 15), 2022-09-01  <- Only covers until Sept 2022!
```

**Action 4-8:** ✗ Correct logic, wrong data

```
# Date parsing and filtering (correct implementation)
df['start_dt'] = pd.to_datetime({'year': df['start_year'],
    'month': pd.to_datetime(df['start_month'].str.strip(), format='%b').dt.month,
    'day': df['start_date']})
window = df[df['start_dt'].between('2021-05-01', '2023-06-26')]
print(len(window))  # 70 competitions

# Keyword regex matching (correct patterns)
regex = re.compile(r"\bdeepfakes?\b|\bgpt\b|\bllms?\b|\bgans?\b|...",
                   re.IGNORECASE)
haystack = window['comp_name'] + '␣' + window['Tag'] + '␣' + window['desc']
matched = haystack.apply(lambda s: bool(regex.search(s)))
print(matched.sum())  # Returns 2 (WRONG due to incomplete dataset)
```

- - - - - - - - - - - - - - - - - - - - - - - - - - - - - - - - - - - - - - - - - - - - - - - - - - - - - - -

**Failure Reasons:**

- **Misleading keyword:** Task mentions "competitions" → agent literally matches dataset name `kaggles-all-completed-competition-dataset`
- **Data coverage gap:** Selected dataset only extends to Sept 2022, missing 9 months of requested range (until June 2023)
- **Missed context:** Should use **meta-kaggle** dataset which contains up-to-date competition metadata

**Correct Approach:**

```
# Should have used meta_kaggle with full coverage
$ python -c "df=pd.read_csv('meta-kaggle/Competitions.csv'); df['EnabledDate']=pd.
    to_datetime(df['EnabledDate']); result=df[df['EnabledDate'].between
    ('2021-05-01','2023-06-26')]; print(len(result[result['Title'].str.contains('gpt|
    llm|gan|deepfake', case=False)]))"
# Output: 10 (correct answer)
```

**Failure Case 2: Data Processing Semantic Error**

---

### Failure Case

**Task:** Determine integer factor of BTX level change (2017-2018) and O3 trend (2015-2020).
**Dataset:** air-quality-data-in-india (city_day.csv)
**Expected Answer:** 3; Stable    **Agent Answer:** 2; Stable    ✗**PARTIAL**

**Key Actions (40 messages):**
**Action 1-3:** ✓ Dataset discovery and exploration

```
$ ls -1  # Lists 30+ datasets
$ python -c "df=pd.read_csv('air-quality-data-in-india/source/city_day.csv'); print(
    df.columns.tolist())"
# Output: ['City', 'Date', 'PM2.5', ..., 'Benzene', 'Toluene', 'Xylene', ...]
```

**Action 4:** ✓ Missing data analysis

```
$ python -c "for col in ['Benzene','Toluene','Xylene']: print(f'{col}: {df[col].isna
    ().mean():.1%}')"
# Output: Benzene: 24.7%, Toluene: 24.7%, Xylene: 61.3%  <- High missing!
```

**Action 5:** ✗ **BTX calculation semantic error** (critical)

```
# Agent's Implementation - WRONG
df["BTX"] = df[["Benzene", "Toluene", "Xylene"]].sum(axis=1, min_count=1)
# Problem: min_count=1 treats NaN as 0, not as missing measurement!
# Example: Benzene=6.8, Toluene=16.4, Xylene=NaN -> BTX=23.2 (should be NaN)
```

**Action 6-8:** ✓ Yearly aggregation (propagates error)

```
$ python -c "yearly=df.groupby('Year').agg(avg_BTX=('BTX','mean'), avg_O3=('O3','mean
    '));print(yearly.loc[2017:2018])"
# Output: 2017: BTX=7.22 (biased low), 2018: BTX=14.64 (biased low)
# Factor: 14.64/7.22 = 2.03 -> round to 2 (WRONG, should be 3)
```

**Action 9-11:** ✓ O3 trend analysis (correct)

```
# Spearman correlation for monotonic trend
subset = yearly.loc[2015:2020, "avg_O3"]
positions = pd.Series(range(len(subset)))
ranks = subset.rank(method="average")
rho = positions.corr(ranks)  # rho = 0.143
trend = 'Stable' if abs(rho) < 0.5 else ...  # Correct!
```

- - - - - - - - - - - - - - - - - - - - - - - - - - - - - - - - - - - - - - - - - - - - - - - - - - - - - - - - - - -

**Failure Reasons:**
- **Semantic error:** min_count=1 parameter treats missing values as zero
- **Domain knowledge gap:** Unmeasured pollutant concentration $\neq$ zero concentration
- **Correct implementation:** Should use df['BTX'] = df['Benzene'] + df['Toluene'] + df['Xylene'] (NaN propagates naturally)

**Impact:** Biased BTX estimates $\rightarrow$ wrong factor (2 vs. 3). O3 analysis unaffected
**Reference Implementation:**

```
# Correct: NaN propagation (any missing component -> entire BTX is NaN)
city_day['BTX'] = city_day['Benzene'] + city_day['Toluene'] + city_day['Xylene']
yearly = city_day.groupby('Year')['BTX'].mean()
factor = int(round(yearly.loc[2018] / yearly.loc[2017]))  # 43.18/15.47 = 2.79 -> 3
```

## G. Prompts

This section presents the key system prompts used in our benchmark construction pipeline: anchor extraction, question refinement, error analysis, adversarial evolution, and agent execution.

**Prompt 1: Solution Anchor Identification**

---

### Identify Solution Anchors from Notebooks

**Role:** Expert data analyst identifying solution anchors with precise numerical values.
**Input:** {notebook_content}
**Task:** Extract cell groups forming complete analytical insights (solution anchors):

1. **Question Cells**: Markdown cells with analytical questions/themes

2. **Answer Cells**: Markdown cells with conclusions + **specific numbers**

3. **Code Cells**: Core analytical code (exclude loading/preprocessing)

**Quality Requirements:**

- Numerical precision (counts, percentages, statistics)

- Exclude ML metrics (accuracy, F1-score)

- Focus on EDA/statistical/descriptive analytics

- Completeness (all three components required)

**Good Example:** *[Example with Cell 15-18 showing question→code→answer flow]*
**What to REJECT:**

- Simple stats without context ("Dataset has 1000 rows")

- Model results ("Accuracy is 95%")

- Answers without numerical values

**Output Format:**

```
{
  "insights": [
    {"question_cells": [15], "answer_cells": [18], "code_cells": [16, 17]}
  ],
  "total_insights": 1
}
```

**Principle:** Quality over quantity - extract only insights meeting ALL criteria.

---

**Prompt 2: Question Formulation**

---

## Formulate Natural Language Questions from Anchors

**Role:** Expert data analyst extracting QA pairs from analysis notebooks.
**Input:** {`notebook_prefix, question_cells, answer_cells, code_cells`}
**Four Extraction Requirements:**

1. **Extract and Refine the Question**

   - Specific: Ask for concrete, quantifiable information
   - Unambiguous: Single correct answer
   - Self-contained: Include all context (no pronouns)
   - Dataset-agnostic: NO dataset names or file names

2. **Extract and Refine the Answer**

   - Precise: Only core numerical values/facts
   - Concise: Remove explanatory text
   - Structured: Use semicolons to separate values
   - Exact: No approximations ($>$, $<$, $\sim$, about, etc.)

3. **Identify Required Datasets**

   - Format: `dataset-slug/filename`
   - Only include datasets actually loaded in code

4. **Quality Control - Should This QA Be Kept?**

   - REJECT if: imprecise ($>$, $\sim$), vague, multiple interpretations, ML metrics
   - KEEP if: exact values, single interpretation, verifiable from code

**Good Example:**

```
# Question: "What is the maximum number of respondents from
#            Kaggle-anonymized countries, and what percentage?"
# Answer: "36; 0.61%"
# Guidelines: "Format: count; percentage (2 decimals).
#              If not applicable: 'Not Applicable'."
```

**Bad Example:**

```
# Question: "How does it compare to others?" (uses pronouns)
# Answer: "Approximately 30%" (contains approximation)
```

**Output:**

```
{
  "should_keep": true,
  "refined_question": "...",
  "refined_answer": "...",
  "answer_guidelines": "...",
  "required_datasets": ["dataset-slug/file.csv"]
}
```

**Principle:** Be strict - when in doubt, REJECT. Only keep exact, verifiable QA pairs.

**Prompt 3: Adversarial Verification**

## Verify Task Quality Through Error Analysis

**Role:** Data science expert analyzing incorrect model answers.

**Input:** {question, expected_answer, model_answer, reference_code, model_code, trajectory_summary}

**Two-Step Analysis:**

**Step 1: Determine Root Cause Source**

- **A. Model Limitations** ($\rightarrow$ MODEL_FAILURE, skip analysis):
    - Gave up, runtime errors, incorrect logic
    - Failed to access symlinked data files
    - Ignored provided information

- **B. Dataset/Question Issues** ($\rightarrow$ classify TYPE_1/2/3):
    - Question lacks necessary information
    - Answer/guidelines ambiguous
    - Multiple datasets match requirements

**Step 2: Error Type Classification (if Step 1 = B)**

```
# TYPE_1_INFO_MISSING: Question lacks necessary info
# - Column rename not mentioned
# - Preprocessing steps not specified
# Fix: Modify question (if standard) OR answer/code (if non-standard)

# TYPE_2_ANSWER_NOT_UNIQUE: Multiple valid interpretations
# - Calculation method varies reasonably
# Fix: Modify answer/code to accept model's interpretation

# TYPE_3_DATASET_AMBIGUOUS: Multiple datasets satisfy requirements
# - Model used different dataset matching requirements
# Fix: Modify question to add distinguishing info
```

**Critical Constraints for Fixes:**

1. NEVER mention dataset names explicitly

2. Add only necessary missing information

3. Answers must be definite numerical values

4. Minimal code changes, preserve absolute paths

**Output:** Error classification, root cause analysis, proposed fixes (question/answer/code modifications).

**Prompt 4: Adversarial Evolution**

## Evolve Tasks Toward Maximal Difficulty

**Role:** Data science education expert using an adversarial approach to increase difficulty.
**Input:** {`question, answer, answer_guidelines, reference_code`}
**Goal (Adversarial Thinking):**
Balance competing objectives:

1. **Make harder** (Discriminator): Remove explicit hints

2. **Maintain uniqueness** (Generator): Ensure single answer

**Three-Task Analysis:**
**Task 1: Identify Overly Explicit Information**

- Explicit dataset keywords ("credit card", "housing")

- Specific column names making identification trivial

- Unnecessary analysis method details

- Numerical hints inferrable from data

**Task 2: Propose Simplifications**

```
# Strategy 1: Replace specific with generic
# "credit card application" -> "application data"

# Strategy 2: Fuzzy column references
# "AMT_INCOME_TOTAL" -> "total income column"

# Strategy 3: Omit standard details
# Remove: "calculate percentile", "apply IQR method"

# Strategy 4: Keep essential constraints
# Preserve: thresholds affecting unique answer
```

**Task 3: Verify Answer Uniqueness**

- Can simplified question still lead to unique answer?

- Are multiple datasets satisfied?

- Would analyst arrive at same answer?

**Output:**

```
{
  "status": "simplified",
  "simplified_question": "...",
  "simplifications_made": [
    {"type": "dataset_keyword_removed", "reasoning": "..."}
  ],
  "answer_uniqueness_analysis": {"still_unique": true, "confidence": "high"}
}
```

**Principle:** Conservative - if unsure about uniqueness, don't simplify.

**Agent System Prompt: Task Execution**

## OpenHands Agent Instructions

**Role:** AI assistant that interacts with computer to solve tasks through code execution.
**Environment:** Working directory with all necessary data and tools.
**Primary Responsibilities:**

- Execute commands, modify code, solve technical problems

- Prioritize quality over speed, be thorough and methodical

- If user asks "why", answer the question (don't try to fix)

**Key Guidelines (6 Categories):**
**1. Efficiency**

- Combine multiple actions (e.g., multiple bash commands into one)

- Use efficient tools: `find`, `grep`, `sed` with filters

**2. File System**

- Explore file system before working (don't assume paths)

- Edit files directly, NOT create new versions with suffixes

- Delete temporary files after testing

- NEVER create: `file_test.py`, `file_fix.py` variants

**3. Code Quality**

- Write clean, efficient code with minimal comments

- Make minimal changes to solve problems

- Understand codebase through exploration first

- Place all imports at top of file

**4. Version Control**

- Use existing git credentials or default: `openhands@all-hands.dev`

- Exercise caution: NO dangerous changes (push to main, delete repos) unless asked

- Use `git commit -a` when possible, check `.gitignore`

**5. Problem-Solving Workflow**

```
1. EXPLORATION: Understand context before solutions
2. ANALYSIS: Consider multiple approaches
3. TESTING: Create tests for bugs/features (skip for docs/configs)
4. IMPLEMENTATION: Minimal, focused changes
5. VERIFICATION: Test thoroughly if environment set up
```

## OpenHands Agent Instructions (continued)

**6. Security (3 Levels)**

- **OK without consent**: Download/run code from user-specified repos, install popular packages

- **Require consent**: Upload code/keys to non-original locations

- **Never do**: Illegal activities, circumvent security, mine cryptocurrency

**Security Risk Assessment:**

```
# LOW: Read-only actions (view files, calculations)
# MEDIUM: Project-scoped edits (modify files, run tests, install packages)
# HIGH: System-level ops (sudo, global installs, delete critical files)
# Rule: Always HIGH if sensitive data leaves environment
```

**Tools Available:** `terminal`, `file_editor`, `task_tracker`, `finish`, `think`

**Response Format:**
For each input, analyze and take action:

1. Understand task and environment

2. Choose appropriate tool and parameters

3. Execute action with proper `security_risk` assessment

**Example:**

```
User: "Fix the bug in data_processor.py"
1. Exploration: terminal(command="find . -name 'data_processor.py'",
                         security_risk="LOW")
2. Analysis: file_editor(command="view", path="src/data_processor.py",
                         security_risk="LOW")
3. Implementation: file_editor(command="str_replace", ...,
                               security_risk="MEDIUM")
4. Verification: terminal(command="pytest tests/test_processor.py",
                          security_risk="MEDIUM")
```

# H. Limitations and Future Directions

**Current Scope and Domain Bias**   CoDA-Bench is constructed from the Kaggle ecosystem, which introduces several limitations. First, Kaggle tasks emphasize exploratory data analysis and predictive modeling, which may not fully represent other data-intensive domains such as ETL pipelines or real-time analytics. Second, Kaggle datasets are typically curated and well-documented, whereas real-world data repositories often contain messier, less structured data. Third, Kaggle workflows are single-user and notebook-based, whereas enterprise workflows may involve multi-user collaboration and production pipelines.

Despite these limitations, CoDA-Bench captures a core challenge that generalizes beyond Kaggle: discovering relevant data in large, noisy environments before performing analysis. Our construction pipeline is domain-agnostic and relies on three transferable principles: (1) community construction via co-occurrence analysis, (2) solution-based back-construction with verifiable outputs, and (3) adversarial evolution via model-based difficulty control. Leveraging these principles, we plan to extend CoDA-Bench to broader data-intensive scenarios, including ETL workflows in enterprise data lakes, code repositories on GitHub with associated datasets, and scientific data analysis pipelines. In future work, we will explore whether performance on CoDA-Bench correlates with performance in these broader domains.

**Contamination Risk.**   Kaggle datasets are publicly available and may appear in model training corpora. While our tasks require precise computation on actual data rather than recall, models may benefit from familiarity with dataset schemas or common patterns. To address this, we plan to: (1) periodically release new benchmark versions based on recent Kaggle datasets and analyses that post-date model training cutoffs, and (2) expand data sources to include private-domain datasets from enterprise and research institutions that are not publicly accessible.

