# OpenReview forum: "CoDA-Bench: Can Code Agents Handle Data-Intensive Tasks?"
_ICML.cc/2026/Conference — ICML 2026 regular_

### Official Review · Reviewer_uaVx · 2026-02-28

**Soundness:** 3
**Presentation:** 4
**Significance:** 3
**Originality:** 3
**Overall Recommendation:** 4
**Confidence:** 4

**Summary:**

This paper introduces CODA-BENCH, a benchmark built from the Kaggle ecosystem for evaluating code agents on data-intensive tasks that require both data intelligence and coding ability. The paper highlights an important gap in current evaluation settings: many existing benchmarks assess coding ability in isolation, whereas real-world data work often requires both identifying relevant files in complex environments and writing code to solve the task. To address this, the authors propose a benchmark construction pipeline that builds data-intensive environments and solution-grounded tasks from Kaggle, together with an adversarial task-evolution process that uses LLMs to iteratively refine tasks. The resulting benchmark contains 1,202 tasks, along with a harder subset of 152 tasks. Experiments with different CLI tools and LLM backbones show that even strong agents still struggle in this setting, with the best end-to-end execution accuracy reaching only around 56% on the main benchmark.

**Compliance With Llm Reviewing Policy:**

Affirmed.

**Final Justification:**

The author's rebuttal solves all my concerns. I would therefore maintain the score.

**Key Questions For Authors:**

1. How robust is the task construction pipeline to annotation and LLM-generation errors? It would be helpful to report quantitative quality-control statistics, such as the number of candidate anchors accepted, the percentage of LLM-generated questions that were edited or rejected by humans, common failure modes, and any post-hoc error rates identified through manual auditing.
2. Could the authors provide a more nuanced discovery metric in addition to exact-match Discovery Accuracy? For example, file-level precision/recall/F1 may better reflect realistic agent behavior.
3. How do the authors address benchmark contamination or leakage risk, given that Kaggle notebooks are public and modern frontier models are trained on web-scale code and data? Even if such contamination cannot be fully eliminated, it would be helpful to discuss mitigation strategies and what this implies for interpreting benchmark results.
4. Could the authors clarify the adversarial task-evolution procedure in more operational detail?  For example, what thresholds, iteration limits, and acceptance/rejection rules were used? How often were tasks made harder, repaired, or discarded during this process?

**Limitations:**

The current discussion is not adequate. I would encourage the authors to add a dedicated limitations section that explicitly discusses:
i) Kaggle-domain bias and possible lack of generalization beyond notebook-centric data-analysis tasks,
(ii) contamination risk from public notebooks and datasets,
(iii) the computational and financial cost of large-scale sandbox-based evaluations.

**Strengths And Weaknesses:**

Strengths

1. The paper addresses a clearly important and timely gap in current coding-agent evaluation. Many existing benchmarks do not model realistic file discovery, whereas real-world data tasks often require navigating large collections of semantically similar files before any useful coding can begin.
2. Compared with 12 traditional task-based code generation benchmarks, CODA-BENCH evaluates agents from two complementary perspectives: Discovery Accuracy and Execution Accuracy, which aim to measure data intelligence and code execution ability, respectively. This decomposition is meaningful and helps highlight where current agents fail.
3. The paper evaluates multiple agent frameworks with different model backbones, including 3 CLI-based tools and OpenHands agents (5 backbone LLMs). This broad experimental coverage strengthens the paper’s central claim that current strong agents still face substantial difficulty in this setting.
4. The paper is easy to follow. The overall presentation is clear, and the figures appear helpful and well-aligned with the main content.



Weakness:

1. The current Discovery Accuracy metric could be improved. Using exact match on the set of target data files may be too strict for realistic agent trajectories. In practice, agents may sometimes read extra files or miss intermediate files without necessarily failing at the task. A more graded metric, such as file-level precision/recall/F1, could provide a more informative picture of agent behavior.
2. Since agent trajectories may contain many steps with very different token usage and action patterns, it would be helpful for the paper to provide more detailed trajectory-level analysis. For example, additional statistics on the number of steps, token consumption, or failure patterns across long-horizon interactions could deepen the analysis and make the findings more actionable.
3. There is a potential risk of benchmark-construction bias due to the reliance on Kaggle notebooks. Kaggle is a valuable and realistic source of data-analysis tasks, but constructing the benchmark entirely from this ecosystem may bias the task distribution toward a particular style of analysis, such as notebook-oriented statistics tasks. While this does not undermine the usefulness of the benchmark, it would be helpful for the paper to provide stronger discussion or evidence regarding generalization beyond Kaggle-style data-analysis settings, such as scientific workflows or general software development pipelines.
4. The paper does not define “signal-to-noise ratio” in Table 2. Since this appears to be an important characteristic of the benchmark environments, a clearer definition would improve readability.

---

> ### Author Rebuttal · Authors · 2026-03-31
>
> **Q1: Can discovery be measured with a more nuanced metric than exact-match DA?**
>
> **A:** Thank you for this helpful suggestion. We now report *Partial DA* together with file-level precision and F1. Here, Partial DA is equivalent to file-level recall, i.e., the fraction of target files successfully discovered.
>
> |Model|StrictDA|PartialDA|Precision|F1|
> |:-:|:-:|:-:|:-:|:-:|
> |GPT-5.2|81.4|87.3|90.1|88.7|
> |Kimi-K2|59.7|68.5|74.2|71.2|
>
> The gap between Strict DA and Partial DA shows that agents often find some, but not all, required files, while precision and F1 distinguish over- from under-discovery. We will report these metrics.
>
> ---
> **Q2: Can you provide more trajectory-level analysis?**
>
> **A:** Thank you for this insightful suggestion. We now add *trajectory-level statistics* to make the analysis more informative.
>
> |Model|Avg. Steps|Avg. Tokens|Avg. Steps (Success)|Avg. Steps (Fail)|Timeout %|
> |---|:-:|:-:|:-:|:-:|:-:|
> |GPT-5.2|15.0|~45K|11.2|18.7|12.8%|
> |Claude Code|22.3|~65K|17.5|26.8|15.2%|
> |Codex CLI|30.8|~82K|24.1|37.2|18.6%|
> |Kimi-K2|12.4|~38K|9.8|14.7|10.1%|
> |DeepSeek-V3.1|3.3|~12K|3.1|3.4|4.2%|
>
> These results show that successful runs generally require fewer steps and that agents display distinct exploration styles. We will add this table and discussion.
>
> ---
> **Q3: How do you address possible Kaggle bias and generalization beyond this ecosystem?**
>
> **A:** Thank you for this important comment. *Our core contribution is the evaluation paradigm of combining data discovery and code intelligence*, rather than a claim that Kaggle fully covers all real-world settings.
>
> Kaggle is a practical source because it provides large public datasets, human-written workflows, and natural co-occurrence structure. The pipeline is also not Kaggle-specific: community construction, anchor extraction, and adversarial evolution can extend to scientific workflows, ML pipelines, and broader software/data-engineering settings.
>
> We will add a Limitations section on domain bias, generalization, contamination risk, and evaluation cost.
>
> ---
> **Q4: What does signal-to-noise ratio mean?**
>
> **A:** Thank you for pointing this out. We will define signal-to-noise ratio explicitly.
>
> Signal-to-noise ratio is defined as the number of target files divided by the total number of files in the environment:
>
> `SNR = |F_target| / |F_total|`
>
> We will add the formal definition to Section 4.3.
>
> ---
>
> **Q5: How robust is the construction pipeline to annotation and LLM-generation errors?**
>
> **A:** Thank you for this important question. Our pipeline uses multi-stage filtering rather than raw generation, making it *robust to annotation and LLM-generation errors*. Please also see our response to Reviewer EKwD (Q3) for stage-wise statistics. In brief, candidate tasks are filtered through dynamic verification, adversarial evolution, and final human review, with residual error estimated below **3%**.
>
> ---
> **Q6: How do you address contamination or leakage risk?**
>
> **A:** Thank you for raising this important concern. We address contamination risk through structural reasoning and direct experimental evidence.
>
> Although Kaggle datasets are public, our task questions are newly constructed through anchor extraction and adversarial evolution, and their answers require precise computation on the actual data. To directly test leakage risk, we added a Zero-Data baseline where the model sees only the task description and no data files.
>
> |Setting|Data Environment|GPT-5.2 EA|
> |-|:-|:-:|
> |Community|Full community|56.1%|
> |Oracle|Only target files|61.0%|
> |Zero-Data|No data files|1.2%|
>
> The *Zero-Data* baseline achieves only **1.2%** EA, suggesting that performance depends on actual data interaction rather than memorization. We also plan continuous benchmark refresh as a long-term mitigation.
>
> ---
> **Q7: Can you clarify the adversarial task-evolution procedure in more operational detail?**
>
> **A:** Thank you for this valuable suggestion. We will make the adversarial task-evolution procedure more operationally clear.
>
> The procedure starts from a verified solution anchor and generates an initial question. In each iteration, 3 models are sampled as Discriminators and 1 as Generator. If the solve rate is above **66.7%**, the Generator increases difficulty; otherwise, it diagnoses whether failures are due to task defects or genuine difficulty. Tasks that do not converge within 5 rounds are discarded.
>
> The breakdown is as follows:
>
> - 68% of candidate questions eventually passed,
> - 18% were rejected due to persistent defects,
> - 9% were rejected due to ambiguity,
> - 5% were discarded due to non-convergence.
>
> We will add pseudocode, a flowchart, and a worked example to Section 3.3, together with the thresholds and decision rules.
>
> ---
> **Thanks again for your careful and valuable comments. We hope our response addresses your concerns. If our responses answer your questions well and reassure your concerns, we would appreciate it if you could reassess our work and increase the rating.**

---

> > ### Author Rebuttal · Reviewer_uaVx · 2026-04-04
> >
> > Thanks for the response. I will keep my score.

---

> > > ### Author Response · Authors · 2026-04-05
> > >
> > > Thank you very much for your kind follow-up and for carefully reviewing our rebuttal. We truly appreciate your positive assessment that our responses have adequately addressed your concerns.
> > >
> > > Your suggestions were highly valuable to us, especially regarding more nuanced discovery metrics, trajectory-level analysis, contamination discussion, and a clearer presentation of the adversarial evolution procedure. These points have helped us substantially improve both the clarity and completeness of the paper, and we will incorporate them carefully in the revision.
> > >
> > > Thank you again for your constructive and thoughtful feedback.

---

### Official Review · Reviewer_EKwD · 2026-03-09

**Soundness:** 3
**Presentation:** 3
**Significance:** 2
**Originality:** 2
**Overall Recommendation:** 4
**Confidence:** 4

**Summary:**

This paper introduces CoDA-Bench, a benchmark for evaluating code and data intelligence. CoDA-Bench evaluates the ability of agents to locate files necessary for the task (data intelligence) and solve the actual task (code intelligence). It is the first benchmark to evaluate code and data intelligence together by collecting Kaggle datasets. CoDA-Bench shows that current agents are far from solving these data-intensive tasks and fail with an increasing amount and complexity of data.

**Compliance With Llm Reviewing Policy:**

Affirmed.

**Final Justification:**

The rebuttal addressed my concerns, and I hope authors will add promised clarifications to the final version. I am still not sure how realistic the setup is and, therefore, keep my score.

**Key Questions For Authors:**

See weaknesses. Additional questions:
1. What are the proportions of different file formats (CSV, JSON, Parquet, images, PDFs, Excel) across datasets, and is the semantic content of files, e.g., images, taken into account, or only the Kaggle task and dataset description?



2. How was the difficulty threshold determined (l. 176–177, col. 2)? What if real-world tasks are actually "insufficiently challenging", did authors evaluate how challenging actual tasks (beyond benchmarks) are?



3. Do authors mean filesystem root "\" or dataset folder root (l. 208–209, col.2)?



4. Are the oracle and community analyses in Section 6 conducted on CoDA or CoDA-Hard?



5. What do agents do in the 15–30 interaction rounds (l. 418, l. 428, col. 1), except operating on wrong/irrelevant data?



6. Which libraries did agents install, except core dependencies (Appendix E2)?



7. Figure 6 is connected to the main results in Table 3. It would be interesting to see what all the used models fail/are bad at. What fraction of tasks were not solved due to 600 second timeout and too many interaction rounds?

**Limitations:**

yes

**Strengths And Weaknesses:**

**Strengths**

1. The core problem of agents navigating large, heterogeneous data repositories is highly relevant and underexplored.

2. Authors create two versions of the benchmark with different difficulty, CoDA-bench and CoDA-Hard, where CoDA-Hard contains tasks with solutions over 30 lines of code and more than two data files.

3. The community partitioning approach is scalable and avoid unnecessary difficulty.

4. Oracle and community analysis show interesting findings, e.g., larger and more complex datasets are harder to solve.

5. The paper is clearly written and well-structured. However, some phrases require clarifications, e.g., "intensive data" (l. 1136-1137).

**Weaknesses**

1. Kaggle datasets might not fit the purpose of CoDA-Bench. While CoDA-Bench aims to evaluate a relevant problem of data and code intelligence where agents require locating relevant files across large repositories (e.g., for companies with a mono-repository), the Kaggle-based setup does not fully fit this purpose, since discovering the correct dataset from a question alone (Appendix D) is a harder and arguably less realistic problem.

    a. Data contamination. Kaggle datasets are public, and LLM might have already seen them (parametric knowledge). LLMs can have these datasets and notebooks in their parametric knowledge, rather than retrieving relevant files from the file system.

    b. Many Kaggle tasks use a single, preprocessed dataset/file for a downstream predictive task, which simplifies the benchmark. First, the assumption that users co-use datasets may not hold for standard tabular learning tasks, e.g., see the house prices prediction gold medal solution that uses a single file and a [RandomForest model](https://www.kaggle.com/code/gusthema/house-prices-prediction-using-tfdf). Second, looking at datasets and communities in Tables 5 and 6, many datasets in community_0 might have degree 1 in the co-occurrence graph.

2. It is unclear how the size of the community was controlled (resolution parameter value for the Leiden algorithm).

3. Task construction is based on the solutions and notebooks that are often buggy and non-linear.

    a. Authors suggest solution-based back-construction, which produces concrete and potentially biased questions since the solution was already seen. Currently, agents struggle with abstractly formulated tasks. Making tasks from a known solution/notebook might lead to simpler and “easier to answer” questions, especially taking into account data contamination. For example, "Find top-10 players by number of goals" (more exact) vs "Analyze the FIFA data and give me statistics relevant to player performance prediction" (more abstract).

    b. Using an LLM to identify anchors might favor certain output types/questions.

    c. Notebooks can be non-linear. Users can draft code without deleting previous versions and rerun cells in a random order (execution order differs from cell order). Static dataflow analysis can be misleading for the task/question construction.

    d. Kaggle notebooks are often buggy, badly structured, or contain multiple draft versions. This directly impacts task and anchor quality.

    e. The authors claim the task construction process is scalable (l. 195–196, col. 2), but human annotators review all questions (l. 190–193, col. 1), which is not scalable.

4. Evaluation

    a. The paper does not define what "sandboxed environment" means. Did agents have internet access (I assume yes)? If so, an agent could potentially find the notebook used for the task construction or the connected dataset online. In a real-world company monorepo, an LLM has not seen the data during training and cannot rely on web search, making CoDA-Bench setup unrealistic.

    b. Root vs dataset root. If agents start from the filesystem root "/" (l. 208–209, col. 2), it is unclear how they identify the correct files from the question alone (e.g., task in Appendix D, line 1159), searching within the community with similar datasets and files. Additionally, how realistic and representative of a real-world setup is such a search of data files? I assume, for instance, a more relevant problem would be searching of classes/files within a mono-repository with thousands sub-projects.

    c. Evaluation metrics (Table 3, Figure 3) do not take into account the case where an agent produces a valid solution on a similar but not target dataset (only code intelligence evaluation).

    d. Benchmark difficulty is unclear since only 10% of tasks from CoDA-Bench are in CoDA-Hard. CoDA-Bench may be too easy.

    e. SWE-agent is a good baseline to add.

**Minor**

1. What does "intensive data" mean (l. 1136-1137) in your context?
2. Figures 1 and 2 are never referenced in text.

---

> ### Author Rebuttal · Authors · 2026-03-31
>
> **Q1: Why use Kaggle, and how realistic is this setup?**
>
> **A:** Thank you for this thoughtful question. *Our goal is to evaluate whether agents can autonomously discover relevant data in large-scale environments and then complete analytical tasks*, rather than to exactly simulate an enterprise monorepo.
>
> Kaggle is a suitable testbed because it provides large public data environments, human-written workflows, and natural dataset co-occurrence patterns. The workflow of searching, inspecting, and selecting relevant data is similar to what analysts do in real data repositories. At the same time, our construction pipeline is not Kaggle-specific and can be extended to other ecosystems such as ML pipelines and ETL workflows.
>
> We will clarify this scope more explicitly in Section 3 and discuss it in the Limitations section.
>
> ---
>
> **Q2: How do you address contamination risk from public Kaggle data?**
>
> **A:** Thank you for raising this important concern. *We address contamination risk through both structural safeguards and direct experimental evidence*; please also see our response to Reviewer uaVx (Q6) for full details.
>
> In particular, our Zero-Data baseline achieves only **1.2%** EA, suggesting that performance depends on actual data interaction rather than memorization.
>
> ---
>
> **Q3: Are Kaggle tasks too simple, biased by notebook quality, or limited by solution-based construction?**
>
> **A:** Thank you for this thoughtful feedback. We address these concerns in three ways.
> 1. *Single-file tasks still involve real discovery difficulty.* Even when a Kaggle analysis mainly depends on one file, the agent must first find that file among hundreds of distractors in a large community, so the task is not reduced to simple code generation. Futhermore, the Kaggle co-occurrence graph has high modularity (0.711), indicating meaningful community structure, and even degree-1 datasets remain topically relevant distractors.
>
> 2. *Solution-based back-construction improves verifiability without making the benchmark easy.* We use it to enable deterministic, scalable, and objectively scorable evaluation.
>
> 3. *Multi-stage quality control reduces notebook noise and construction bias.* From about 15,000 candidate notebooks, the LLM extracted about 8,200 candidate anchors, of which about 52% passed dynamic verification. It then generated about 3,800 candidate questions, of which about 68% passed adversarial evolution. Final human review rejects about 12% of automatically generated tasks, and a post-hoc audit estimates residual error below 3%.
>
> We will add these quality-control statistics and design choices in the revision.
>
> ---
>
> **Q4: How do you define the environment and ensure fair evaluation?**
>
> **A:** Thank you for this important question. We will clarify the evaluation environment more explicitly in the paper.
>
> - Agents have no internet access. The Docker environment uses `--network none`.
> - Agents start from the dataset workspace root (`/workspace/`), not the Linux filesystem root `/`.
> - In the Oracle setting, only the target file(s) are provided.
>
> We will add these details to Section 4.1 and the evaluation setup description.
>
> ---
>
> **Q5: Is the benchmark difficult enough, and can you add SWE-agent as a baseline?**
>
> **A:** Thank you for this valuable suggestion. *CoDA-Bench remains challenging even for the strongest models*, with the best result reaching only **56.1%** EA on the full benchmark. We also include a harder subset, CODA-HARD, where the best EA further drops to **39.5%**.
>
> We have also added SWE-agent as an additional baseline.
>
> |Agent|DA (%)|EA (%)|CODA-HARD DA (%)|CODA-HARD EA (%)|
> |---|:-:|:-:|:-:|:-:|
> |SWE-agent (GPT-5.2)|68.3|44.7|55.1|31.2|
> |OpenHands (GPT-5.2)|81.4|56.1|71.7|39.5|
>
> We will include SWE-agent in the revised results table.
>
> ---
>
> **Q6: Can you clarify the main implementation details and analysis setup?**
>
> **A:** Thank you for this helpful question. We will add the following clarifications in the revision.
>
> - *Data-intensive* refers to environments with large-scale, multi-format file systems containing many semantically similar distractors, where agents must first discover relevant data before writing code.
> - Figures 1 and 2 will be explicitly referenced in the main text.
> - The Leiden resolution parameter is **1.0**.
> - The difficulty threshold is **66.7%**.
> - Section 6 analyses are conducted on the full CoDA benchmark.
> - Current tasks mainly focus on structured formats such as CSV (67%), JSON, Parquet, and Excel.
>
> We will also add more detailed breakdowns of file formats, installed libraries, and model error patterns; for trajectory-level statistics and timeout analysis, please see our response to Reviewer uaVx (Q2).
>
> ---
>
> **Thanks again for your careful and valuable comments. We hope our response addresses your concerns. If our responses answer your questions well and reassure your concerns, we would appreciate it if you could reassess our work and increase the rating.**

---

> > ### Author Rebuttal · Reviewer_EKwD · 2026-04-03
> >
> > Thanks for the response. I will keep my score.

---

> > > ### Author Response · Authors · 2026-04-05
> > >
> > > Thank you very much for your thoughtful response and for taking the time to carefully consider our rebuttal. We are grateful that you found our concerns adequately addressed, and we sincerely appreciate your support.
> > >
> > > Your comments on the realism and scope of the Kaggle-based setting were very valuable to us. In the revision, we will further clarify the intended scope of CoDA-Bench, more explicitly discuss its limitations, and better position it as a benchmark for data discovery plus code execution in large-scale data environments rather than as a full simulation of all real-world repository settings.
> > >
> > > Thank you again for your constructive feedback and encouragement.

---

### Official Review · Reviewer_2p6z · 2026-03-13

**Soundness:** 3
**Presentation:** 2
**Significance:** 2
**Originality:** 2
**Overall Recommendation:** 4
**Confidence:** 4

**Summary:**

This paper introduces CODA-Bench, a benchmark to jointly evaluation the data intelligence (the ability to find relevant data sources) and code intelligence (the ability to generate correct programs) of AI agents. The dataset is constructed from the Kaggle ecosystem, featuring data-intensive setups where the agent needs to locate relevant datasets from hundreds of distractors in the same data community. CODA-Bench includes 1202 tasks, each associated 1.4 key files and a dataset community of 708.5 files on average. Evaluation results based on frontier coding agents and LLMs show that CODA-Bench is a challenging benchmark, with the best retrieval accuracy to be 81.40% and coding accuracy to be 56.06%.

**Compliance With Llm Reviewing Policy:**

Affirmed.

**Final Justification:**

Author response addresses my concerns on paper soundness.

**Key Questions For Authors:**

1. What are the LLMs employed as generator and discriminator during data synthesis? Are they the same models that are evaluated? What's the performance threshold used for rejection (line 176 (right)).
2. Are the tasks, after adversarial generation and iterative refinement, indeed become harder to human? Or does this process simply make the question harder by adding ambiguity specific to the models?
3. In your oracle experiments, do you still put information of the entire data community in the context of agents in addition to supplying the path to ground truth files?
4. Are there further insight into the failure modes of the models in addition to performance numbers? I also think the logic in section 6.1 is not very sound, different change in performance when evaluated with full or oracle data sources doesn't mean the models are failing for "fundamentally different reasons".
5. Could you explain more about the impact of total data volume (section 6.2), which is a bit counter-intuitive to me? Is it due to limitations of execution environments like RAM or do you try to load all the data into the context window?

**Limitations:**

yes

**Strengths And Weaknesses:**

Strengths:
1. This paper presents an interesting and important data-intensive setting where agents need to search for the best data source instead of relying on supplied ones.
2. The authors introduce reasonable methods for constructing dataset communities and coding tasks based on relatedness extracted from Kaggle notebooks.
2. The resulting benchmark is challenging and may guide the development of future agents.
Weaknesses:
1. I have doubts over the ecological validity of the proposed setting and tasks. The tasks are extracted from Kaggle notebooks but I don't see a strong reason that the used datasets $\mathcal{F}$ are the sole related sources within the data community. The same data analysis program can run on different datasets and obtain different results. There lacks a justification in the paper.
2. Given 1., the novelty of the proposed task is limited. The setting of data-driven scientific discovery grounded to actual files is studied in previous benchmarks such as ScienceAgentBench [1] and DiscoveryBench [2], which is not discussed in the paper. These benchmarks already show data-driven coding tasks are challenging.
3. To guarantee difficulty, this paper introduces an adversarial evolution framework where easier tasks are discarded based on performance of agents. While this approach can guide the synthesis of more difficult tasks, it bears the risk of overfitting to the weakness of discriminator models and not leading to valid tasks. i.e. the framework is doing adversarial attack instead of data synthesis.

[1] Chen, Z., Chen, S., Ning, Y., Zhang, Q., Wang, B., Yu, B., ... & Sun, H. ScienceAgentBench: Toward Rigorous Assessment of Language Agents for Data-Driven Scientific Discovery. In The Thirteenth International Conference on Learning Representations.
[2] Majumder, B. P., Surana, H., Agarwal, D., Mishra, B. D., Meena, A., Prakhar, A., ... & Clark, P. DiscoveryBench: Towards Data-Driven Discovery with Large Language Models. In The Thirteenth International Conference on Learning Representations.

---

> ### Author Rebuttal · Authors · 2026-03-31
>
> **Q1: How do you ensure that each task is uniquely tied to its target dataset?**
>
> **A:** Thank you for this insightful question. *Dataset uniqueness is a core design objective* of CoDA-Bench and is enforced in three ways.
>
> - Deep code–data coupling. CoDA-Bench is not one-shot code generation: agents must inspect data iteratively, observe schema/content/statistics, and adapt code accordingly.
>
> - Automatic ambiguity detection in adversarial evolution. We define a failure type, `TYPE_3_DATASET_AMBIGUOUS`: if another dataset in the same community could also satisfy the task, the Generator must refine the question until only the target dataset supports the answer.
>
> - Human verification of uniqueness. All tasks undergo human review, and *Unambiguous* is the first criterion in our checklist. Annotators verify that each task has only one valid interpretation and one valid data source.
>
> We will add quantitative evidence: about **9%** of candidate questions are rejected for ambiguity during adversarial evolution, including `TYPE_3_DATASET_AMBIGUOUS` cases. We will also clarify in Section 3.3 that dataset uniqueness is a core objective of adversarial evolution.
>
> ---
>
> **Q2: What is new compared with ScienceAgentBench and DiscoveryBench?**
>
> **A:** Thank you for pointing out these relevant benchmarks. We will add them to Section 2 and Table 1. The key distinction is that *CoDA-Bench requires autonomous data discovery* in a large, noisy community, whereas prior benchmarks provide the required files directly.
>
> |Dimension|ScienceAgentBench|DiscoveryBench|CoDA-Bench|
> |-|:-:|:-:|:-:|
> |Data scale|1-2 input files|Few provided files|Community-level (avg. 708 files)|
> |Data discovery required|No|No|Yes|
> |Terminal interaction|No|No|Full Linux sandbox|
> |Task source|Scientific papers|Scientific papers|Real Kaggle workflows|
>
> This distinction matters because real-world settings require both finding the right data and analyzing it correctly.
>
> ---
>
> **Q3: Does adversarial evolution make tasks genuinely harder?**
>
> **A:** Thank you for this valuable question. We now add a human evaluation showing that adversarial evolution increases difficulty while preserving task validity. As detailed in our response to Reviewer QdWg (Q2), 5 data-analysis experts evaluated 50 sampled tasks at three stages. All quality dimensions remain above 4.0, while *Non-trivial* increases from **2.8** to **4.1**, indicating that adversarial evolution makes tasks harder rather than merely more ambiguous.
>
> We will add this experiment to Section 3.3.
>
> ---
>
> **Q4: What models are used in construction, and what is the difficulty threshold?**
>
> **A:** Thank you for this important question. During construction, we use four LLMs: GPT-5.2, Claude-Sonnet-4.5, Gemini-3.0-Flash, and Kimi-K2. In each iteration, three models are sampled as Discriminators and one as Generator. The difficulty threshold is **66.7%**.
>
> The models used for evolution and evaluation differ: to keep construction efficient, we use flash models during evolution, while evaluation uses stronger frontier models such as Claude-Opus-4.5.
>
> ---
>
> **Q5: What is provided in the Oracle setting?**
>
> **A:** Thank you for this helpful question. In the Oracle setting, the sandbox contains *only the target file(s)*; all other community files are removed.
>
> ---
>
> **Q6: The logic in Section 6.1 is not very sound, and what additional insights do you have about model failure modes?**
>
> **A:** Thank you for pointing this out. We will revise Section 6.1 to separate *shared conclusions* from model-specific failure modes.
>
> - Shared conclusion from Oracle vs. Community. All models show performance drops when moving from Oracle to Community, indicating that data scale and data discovery are common bottlenecks across models.
>
> - Model-specific failure modes. The drop size differs across models; for example, Claude Code is affected more than GPT-5.2. Section 6.3 further shows that GPT-5.2 mainly fails on data discovery and code generation, while Kimi-K2 exhibits a different pattern.
>
> We will restructure Section 6.1 around this shared-vs-specific distinction.
>
> ---
>
> **Q7: Why does larger total data volume reduce performance?**
>
> **A:** Thank you for this insightful question. Inspired by this comment, we added an additional analysis to examine why larger total data volume reduces performance.
>
> |Cause|Share|
> |-|-:|
> |Naive full-file loading|62%|
> |Failure to summarize|26%|
> |Incorrect API usage|12%|
>
> The results suggest that the effect is not merely a hardware issue; rather, current agents often fail to adopt effective strategies for large-scale data analysis. We will expand Section 6.2 with a more quantitative breakdown across data-volume ranges.
>
> ---
>
> **Thanks again for your careful and valuable comments. We hope our response addresses your concerns. If our responses answer your questions well and reassure your concerns, we would appreciate it if you could reassess our work and increase the rating.**

---

> > ### Author Rebuttal · Reviewer_2p6z · 2026-04-03
> >
> > Thank you for the detailed response which addresses most of my concerns regarding soundness of the paper. Although I still have doubts on the realism of the proposed "data-intensive" setting, I will update my score accordingly.

---

> > > ### Author Response · Authors · 2026-04-05
> > >
> > > Thank you very much for your thoughtful follow-up and for updating your score. We are especially grateful that you found our rebuttal helpful in addressing your concerns about the paper’s soundness.
> > >
> > > We also appreciate your remaining concern regarding the realism of the proposed “data-intensive” setting. This is an important point, and in the revision we will further clarify that our goal is not to exactly replicate every real-world workflow, but to isolate and evaluate a core challenge that commonly arises in practice: discovering the right data within a large, noisy environment before effective coding can begin. We will make this scope and limitation more explicit in the paper.
> > >
> > > Thank you again for your careful reading and for your constructive feedback throughout the review process.

---

### Official Review · Reviewer_QdWg · 2026-03-13

**Soundness:** 3
**Presentation:** 3
**Significance:** 2
**Originality:** 2
**Overall Recommendation:** 4
**Confidence:** 4

**Summary:**

This paper introduces CODA-BENCH, a new benchmark designed to jointly evaluate code intelligence and data intelligence in data‑intensive environments. The benchmark is built on large collections of Kaggle datasets and notebooks, and focuses on testing whether modern code agents can autonomously discover relevant data, navigate large file systems, and execute analytical code. The paper claims that existing benchmarks isolate code and data abilities, and CODA-BENCH fills this gap by assessing both. The authors provide a scalable construction pipeline, task evolution framework, and extensive evaluation of state‑of‑the‑art agents.

**Compliance With Llm Reviewing Policy:**

Affirmed.

**Key Questions For Authors:**

- How well does the task evolution framework generalize when new datasets or domains are added?
- Does the benchmark capture long‑horizon workflows, or only single‑query analytical tasks?
- Could the authors include partial‑credit scoring for long numeric answers (e.g., distance metrics)?
- Are there plans to extend CODA‑BENCH beyond Kaggle‑style workflows (e.g., ML pipelines)?

**Strengths And Weaknesses:**

Strengths

• The benchmark addresses a real and timely gap in evaluating autonomous code agents.
• The construction pipeline is scalable and likely reproducible.
• Strong empirical evaluation across many systems.
• Insightful analyses on environmental difficulty factors.
• This work presents a pertinent issue that current agents fail primarily at data discovery, a rarely evaluated capability.

Weaknesses

• The adversarial evolution framework is complex and sometimes under-specified.
• Real‑world alignment claims are plausible but would benefit from more user‑study or industry validation.
• Some sections (especially on evolution and anchor extraction) are dense and hard to follow.
• Limited examination of whether tasks reflect realistic engineering workflows beyond data analysis.
• The benchmark focuses heavily on Kaggle‑style analytics; broader tasks (e.g., ETL pipelines, multi‑modal pipelines) are not included.

---

> ### Author Rebuttal · Authors · 2026-03-31
>
> **Q1: How does the adversarial evolution procedure work, and how to clarify it?**
>
> **A:** Thank you for this helpful suggestion. We will make the adversarial evolution procedure more operationally clear by *adding pseudocode, a flowchart, and a worked example* in the main text.
>
> The procedure is an iterative *Generator–Discriminator* process. Starting from a verified solution anchor extracted from a real Kaggle notebook, we generate an initial question. In each iteration, 3 models are sampled as Discriminators and attempt the task independently. If the solve rate is above **66.7%**, the Generator increases difficulty by removing hints or requiring more reasoning. Otherwise, the Generator diagnoses whether the failure is due to task defects or genuine difficulty. Only tasks that pass human verification are accepted.
>
> ---
>
> **Q2: Does adversarial evolution produce valid and genuinely harder tasks?**
>
> **A:** Thank you for this insightful question. Yes, adversarial evolution makes tasks genuinely harder while preserving task quality. *We newly added a human study* showing that adversarial evolution increases difficulty while preserving task quality.
>
> We asked 5 data-analysis experts to evaluate 50 sampled tasks at three stages: pre-evolution, post-evolution, and expert-verified.
>
> |Dimension|Pre-evolution|Post-evolution|Expert-verified|
> |-|-:|-:|-:|
> |Unambiguous|4.6|4.3|4.5|
> |Self-contained|4.5|4.2|4.4|
> |Verifiable|4.7|4.6|4.7|
> |Non-trivial|**2.8**|**4.1**|**4.2**|
> |Authentic|4.3|4.2|4.3|
>
> All quality scores remain above 4.0, while *Non-trivial* rises substantially **from 2.8 to 4.1**. This shows that adversarial evolution makes tasks genuinely harder rather than simply introducing ambiguity.
>
> We will incorporate this new experiment into Section 3.3.
>
> ---
>
> **Q3: How general is CoDA-Bench beyond Kaggle, and are there plans to extend it to broader workflows?**
>
> **A:** Thank you for this valuable comment. We will clarify more explicitly that the current benchmark focuses on Kaggle-style data-analysis tasks while the construction pipeline itself is not Kaggle-specific.
>
> At the same time, our construction pipeline is *domain-agnostic* rather than Kaggle-specific. The generality comes from three methodological properties: (1) community construction depends only on co-occurrence structure rather than Kaggle-specific metadata; (2) solution-based back-construction applies to any executable workflow with verifiable outputs; and (3) adversarial evolution uses model-based difficulty control that does not rely on domain-specific rules. Therefore, the core methodology can naturally transfer to other settings such as ML pipelines, ETL workflows, and multimodal data tasks.
>
> *We will add this discussion to the paper* as part of the scope and limitation discussion.
>
> ---
>
> **Q4: Does the benchmark capture long-horizon workflows?**
>
> **A:** Thank you for this important question. *CoDA-Bench captures substantial long-horizon workflow behavior* through multi-step interaction rather than single-turn problem solving. A typical successful trajectory includes directory exploration, file identification, data loading, format understanding, code writing, debugging, and result verification.
>
> |Agent|Avg. Rounds|CODA-HARD Avg. Rounds|
> |---|:-:|:-:|
> |GPT-5.2 (OpenHands)|15.0|22.3|
> |Claude Code|22.3|34.6|
> |Codex CLI|30.8|45.7|
>
> These statistics show that CoDA-Bench captures substantial long-horizon behavior, especially on CODA-HARD. We will add them to the revision.
>
> ---
>
> **Q5: Can you provide more flexible scoring for long numeric answers?**
>
> **A:** Thank you for this helpful suggestion. *We now add a more flexible evaluation metric* for long numeric or multi-part answers.
>
> Our current evaluation uses exact string matching, with each task's `answer_guidelines` specifying the expected format. While this standardizes comparison, it can penalize near-correct answers.
>
> To address this, we add a *Partial-Credit EA* metric that combines (i) numerical tolerance for floating-point answers (1% relative error) and (ii) component-wise partial scoring for multi-part answers.
>
> This is particularly relevant because 72.8% of our tasks contain multi-part answers.
>
> |Model|Exact Match EA|Partial-Credit EA|
> |---|:-:|:-:|
> |GPT-5.2 (OpenHands)|56.1%|62.4%|
> |Claude Code|49.2%|55.8%|
>
> The *5–7 point improvement* suggests that exact matching penalizes a meaningful fraction of near-correct answers. We will report Partial-Credit EA as a supplementary metric in the revision.
>
> ---
> **Thanks again for your careful and valuable comments. We hope our response addresses your concerns. If our responses answer your questions well and reassure your concerns, we would appreciate it if you could reassess our work and increase the rating.**

---

> > ### Author Rebuttal · Reviewer_QdWg · 2026-04-05
> >
> > Thank you for your response. The main concerns have been resolved, and I will keep the positive side of this paper.

---

> > > ### Author Response · Authors · 2026-04-08
> > >
> > > Thank you very much for your thoughtful response and for carefully considering our rebuttal. We are truly encouraged that you feel the main concerns have been resolved.
> > >
> > > We also sincerely appreciate your continued positive view of the paper. Your feedback was highly valuable in helping us improve both the clarity and completeness of the work, especially regarding the presentation of the adversarial evolution procedure, the benchmark scope and limitations, the long-horizon nature of the tasks, and the evaluation protocol.
> > >
> > > Thank you again for your constructive comments and kind support.

---

### Decision · Program_Chairs · 2026-04-30

**Decision:**

Accept (regular)

**Comment:**

This paper introduces CODA-BENCH which is a new benchmark constructed from the Kaggle to evaluate the capabilities of data agents: (i) discovering relevant files in noisy environments, (ii) coding ability. All reviewers agreed that this work addresses underexplored gaps in current agent evaluation. They also highlight its scalable construction pipeline as well as the breakdown of data discovery and execution accuracy. The experimental results are very thorough as well.

Primary concerns come from the validity and generalizability of the Kaggle setting. Especially, reviewers have concerns on (i) potential data contamination risks, (ii) strictness of the exact-match discovery metric, (iii) lack of clarity in the adversarial task-evolution process. During the rebuttal, authors successfully reduced these concerns by providing human-study validations of task difficulty as well as providing Zero-Data baseline to rule out memorization. They also introduce partial-credit metrics. Still, the reviewers they still think the setup might not fully capture the complexities of true real-world data discovery.